# Geochemical Markers as a Tool for the Characterization of a Multi-Layer Urban Aquifer: The Case Study of Como (Northern Italy)

**Gilberto Binda [1,2,\*], Francesca Frascoli [1], Davide Spanu [1], Maria F. Ferrario [1], Silvia Terrana [1], Roberto Gambillara [1], Sara Trotta [1], Paula J. Noble [3,4], Franz A. Livio [1], Andrea Pozzi [1] and Alessandro M. Michetti [1,5]**

[1] Dipartimento di Scienza e Alta Tecnologia, Università dell'Insubria, Via Valleggio 11, 22100 Como, Italy; francesca.frascoli@gmail.com (F.F.); davide.spanu@uninsubria.it (D.S.); francesca.ferrario@uninsubria.it (M.F.F.); silvia.terrana@uninsubria.it (S.T.); roberto.gambillara@uninsubria.it (R.G.); strotta@studenti.uninsubria.it (S.T.); franz.livio@uninsubria.it (F.A.L.); andrea.pozzi@uninsubria.it (A.P.); alessandro.michetti@uninsubria.it (A.M.M.)
[2] Norwegian Institute for Water Research (NIVA), Økernveien 94, 0349 Oslo, Norway
[3] Department of Geological Sciences and Engineering, University of Nevada, Reno, NV 89557, USA; noblepj@unr.edu
[4] Global Water Center, University of Nevada, Reno, NV 89557, USA
[5] Istituto Nazionale di Geofisica e Vulcanologia, 80124 Napoli, Italy
**\*** Correspondence: gilberto.binda@uninsubria.it

**Abstract:** The analysis of geochemical markers is a known valid tool to explore the water sources and understand the main factors affecting natural water quality, which are known issues of interest in environmental science. This study reports the application of geochemical markers to characterize and understand the recharge areas of the multi-layer urban aquifer of Como city (northern Italy). This area presents a perfect case study to test geochemical markers: The hydrogeological setting is affected by a layered karst and fractured aquifer in bedrock, a phreatic aquifer hosted in Holocene sediments and connected with a large freshwater body (Lake Como); the aquifers recharge areas and the water geochemistry are unknown; the possible effect of the tectonic setting on water flow was overlooked. In total, 37 water samples were collected including water from two stacked aquifers and surface water to characterize hydrochemical features. Moreover, six sediment samples in the recent palustrine deposits of the Como subsurface were collected from cores and analyzed to understand the main geochemistry and mineralogy of the hosting material. The chemical analyses of water allow to observe a remarkable difference between the shallow and deep aquifers of the study area, highlighting different recharge areas, as well as a different permanence time in the aquifers. The sediment geochemistry, moreover, confirms the differences in trace elements derived from sediment-water interaction in the aquifers. Finally, an anomalous concentration of As in the Como deep aquifer was observed, suggesting the need of more detailed analyses to understand the origin of this element in water. This study confirms the potentials of geochemical markers to characterize main factors affecting natural water quality, as well as a tool for the reconstruction of recharge areas.

**Keywords:** geochemistry; hydrogeology; groundwater; trace elements; water quality; recharge area; sediment

## 1. Introduction

Monitoring of groundwater chemistry and water resource management are important tools in sustaining human and environmental health, as these practices increase understanding of the main processes and stressors affecting water quality [1–3].

Groundwater exploitation for human and industrial use requires the knowledge of the main natural and anthropogenic factors that can affect water quality. Host rock geochemistry and tectonic structural features are natural attributes known to play a major role in groundwater chemical composition and flow path, in addition to the hydrology and geomorphology of the region [4–8]. These attributes, moreover, can greatly affect geochemical background values and can cause high concentrations of potentially toxic elements (PTEs, e.g., As, Ni, Cr, Pb, U, Li), which in turn can negatively affect water quality for human use and consumption [3,9–11].

Geochemical markers are valid tools used to reconstruct groundwater circulation in these contexts. Different chemical variables, such as major ions [12], trace elements [4], and water isotopes [13], can provide helpful insights for the estimation of groundwater circulation and the determination of recharge areas. Their combined use can therefore provide information to develop and test conceptual models of groundwater circulation in complex systems, identify source waters and evaluate potential subsurface interconnections and mixing between local and regional aquifers [5].

Here we identify some geochemical markers from groundwater in the town of Como (northern Italy), which has a complex and multi-layer aquifer hosted in the subsoil of the urban area, to understand the present and future state of water quality. Como is built on a sedimentary basin (elevation ca. 200 m a.s.l.), surrounded by mountain slopes reaching elevations higher than 1000 m a.s.l. The local base level is represented by the surface elevation of Lake Como (ca. 198 m a.s.l.). The physical hydrogeological characteristics (e.g., groundwater level and its seasonal variations, hydraulic connection between the aquifers and Lake Como) have been analyzed in previous studies in order to characterize subsidence phenomena affecting the town of Como and for designing engineering facilities to protect the lakeshore from flooding [14–17]. In contrast, the geochemical features of the aquifer have been largely overlooked and there are some tectonic features which can possibly affect the water flow and chemical composition.

In fact, no data are available on the water chemistry of the Como aquifers, and their main recharge areas are still to be identified. Moreover, this area includes tectonically active structures (i.e., the Gonfolite backthrust [18,19]) and, to the best of our knowledge, their effect on water quality in Como basin has not been investigated yet.

Therefore, this study aims to: (i) characterize the overall geochemistry and water quality of the water in the Como basin aquifers, (ii) construct a conceptual model depicting the main recharge areas and (iii) ascertain the sources of certain PTEs that have previously been found within this complex, multi-layer urban aquifer. These objectives will be reached through the analysis of groundwater, surface water and sediment samples collected in the Como basin and the multifaceted interpretation of geochemical markers. In this way, this work can be applied as a tool for local managing authorities and can serve as a template for understanding the groundwater systems in other urban settings.

## 2. Study Area

The city of Como is located at the southern end of the western branch of Lake Como (Lombardy region, northern Italy), which is a glacial lake in the Italian Southern Alps displaying a characteristic $\lambda$ shape in map view (Figure 1a). The city is located on a small alluvial plain, surrounded by bedrock mountain slopes. No outlets exist in this area, whereas some minor fluvial inlets (i.e., the Cosia and Valduce creeks) cross the urban area.

The region has a temperate climate, with an average annual temperature of 13 °C and a total annual rainfall of 1500 mm, with peaks of precipitation amount in May and in November. Moreover, while the amount of solid precipitation is decreasing in this area

due to recent climate changes, snowpack melting still increases the water runoff in spring [20].

*2.1. Regional Tectonic and Geologic Setting*

The city of Como is included in the Western Southern Alps domain, a notable example of an ancient passive margin later involved into the orogenetic wedge (e.g., [21]). The Mesozoic syn-rift sequence includes pelagic limestones of the Moltrasio Limestone formation (Medolo group, [22,23]), extensively outcropping to the north-east of Como. The involvement of the ancient plate margin in an orogenic wedge caused rapid uplift and erosion of the Alpine belt and is documented by the deep-water clastic sediments of the Upper Oligocene–Middle Miocene Gonfolite Lombarda Group. These sediments include conglomerates with turbiditic sandstones and shales that are incorporated into the N-verging Gonfolite backthrust [19,24].

Previous studies have inferred that the backthrust partially drives, at shallow depths, the upwelling of deep thermal waters from the Triassic Dolostones, outflowing at the Stabio springs, 10 km away from Como [25]. In this area, geochemical analyses were performed on the hot thermal waters, showing a Na-Cl facies and an enrichment in hydrothermally related gases (from isotopic analysis of $CO_2$ and $CH_4$), differently from other springs in the basin [26].

During the Quaternary, glaciers repeatedly occupied what is now the Como basin. During the last glacial maximum (LGM), the Adda glacier occupied a large part of Central Alps and left a strong imprint on the landscape. Glacial deposits in the Como area are widespread up to elevations of ca. 800–900 m a.s.l. [23]. Glacial deposits are abundant at the margin between the Alps and the plain, where the glacier deposited several orders of moraines, each representing a distinct stage.

The study area (Figure 1) can be divided in different sectors with specific geological features relevant to this research:

- To the north-east of Como, a mountain range of carbonate rocks reaches elevations higher than 1000 m a.s.l. and is affected by diffuse karstic features;
- To the west, the Gonfolite Group is composed of marls, at the base, overlaid by a thick sequence of conglomerates (mainly intrusive and metamorphic clasts, with a subordinate carbonate component of <20% [27,28]);
- To the south-east, glacial deposits predominate, lying on relatively impermeable marls of the substratum; their petrographic composition reflects a provenance from the Central Alps (i.e., abundant crystalline and metamorphic clasts), formerly occupied by extensive ice tongues.
- The Cosia stream watershed lies to the east of the city of Como and is mainly composed of carbonate rocks in the upper reaches with the unconsolidated glacial deposits in the lower elevations.

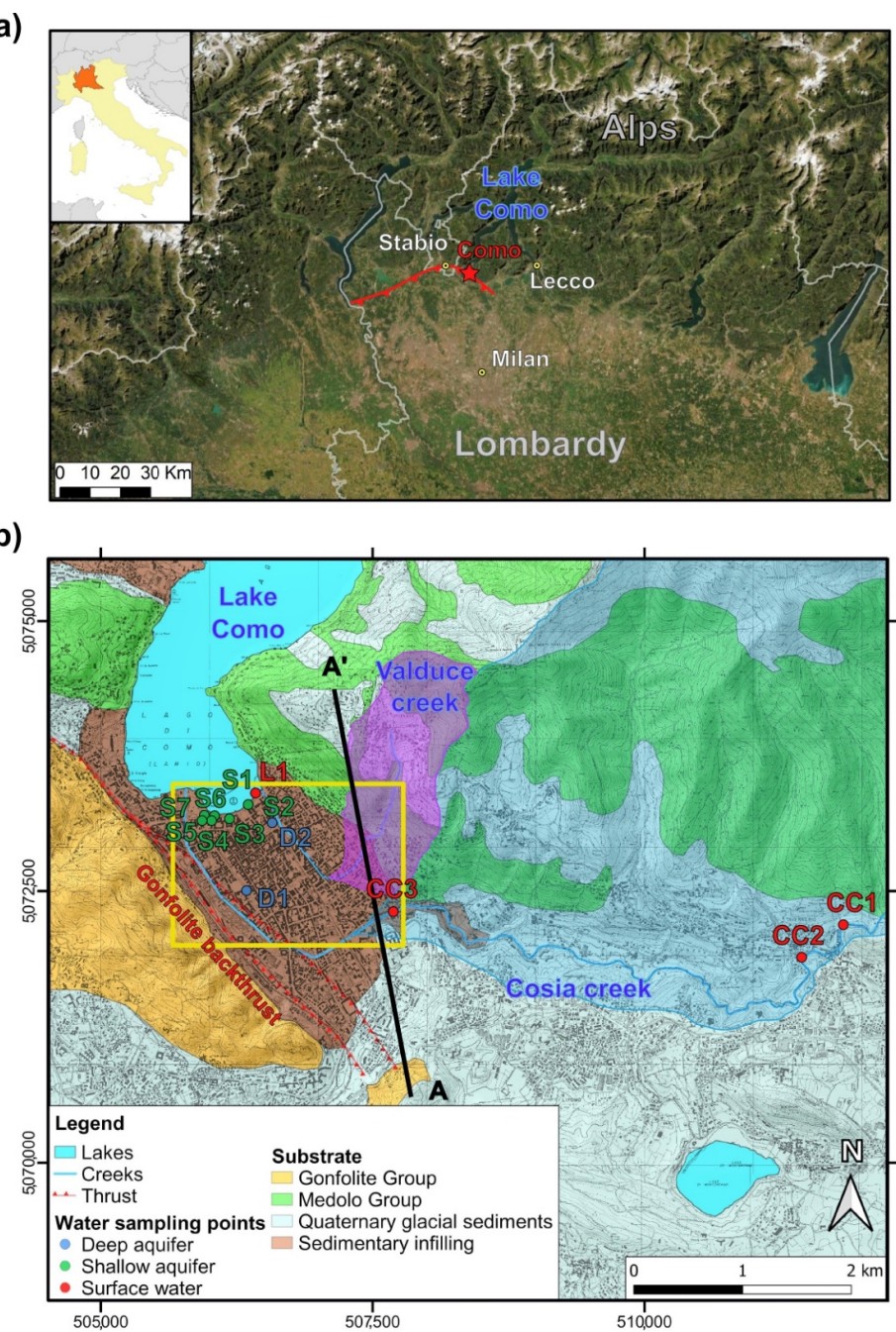

**Figure 1.** (**a**) Location map of the Como urban area, highlighted with a red star; Lombardy region is highlighted in orange in the upper left corner. (**b**) Simplified geologic map of the study area, modified after [14]. Water sampling points are highlighted as well, while the sediment cores are located in point S1 (for coreSI3) and in point D2 (for core SV2), respectively (see Figure 2). The drainage basins of the Valduce creek and Cosia creek are indicated by the shaded purple and light blue polygons, respectively. Moreover, the yellow frame indicates the extension of map in Figure 2a, and the black line indicate the location of section A-A' in Figure 7. The red line in (**a**,**b**) indicates the trace of the Gonfolite backthrust. Topo map: CTR (Carta Tecnica Regionale), provided by Regione Lombardia [29].

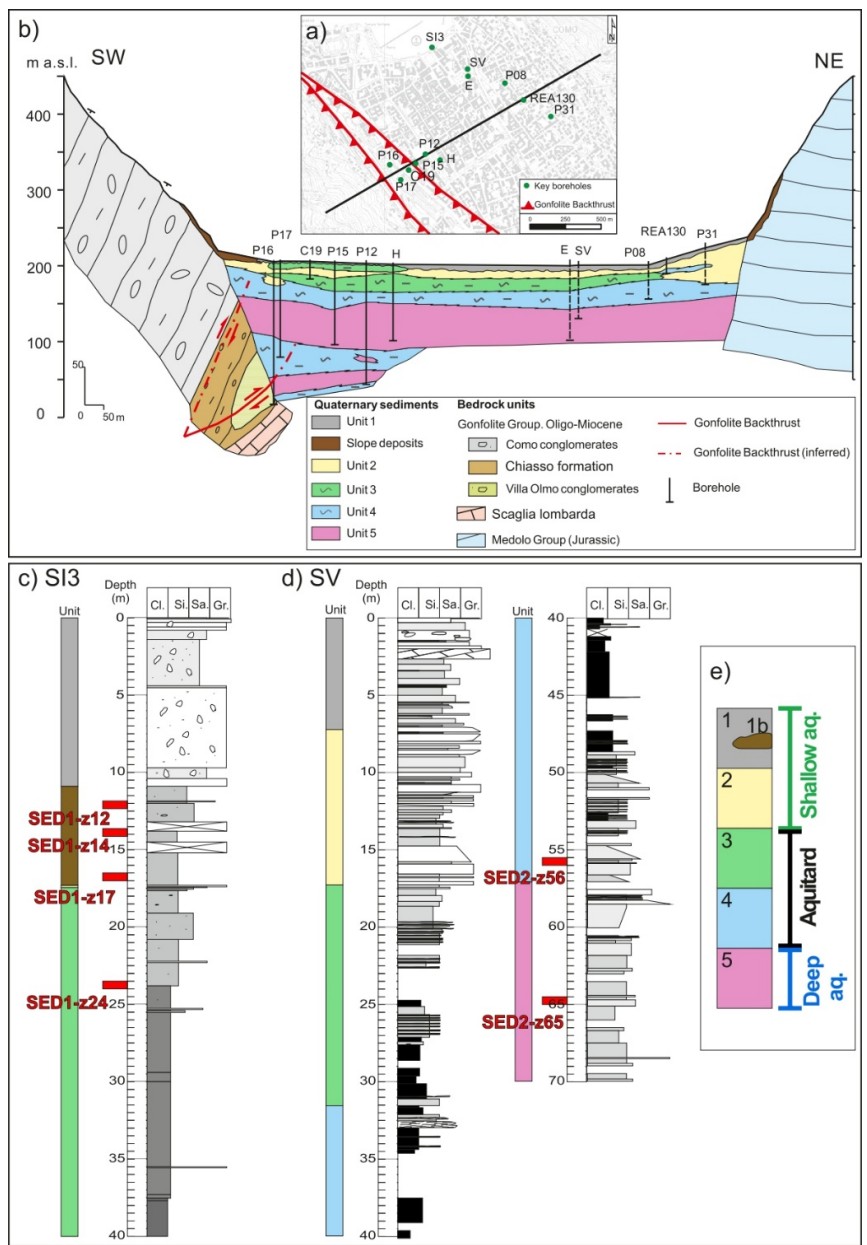

**Figure 2.** (**a**) Location of cores SI3 and SV discussed in the text as well as other cores used to provide subsurface control on the stratigraphy (please refer also to Figure 1b), and the trace of the geological cross-section shown in (**b**). (**b**) Geological cross-section south-west (SW)–north-east (NE) oriented, perpendicular to the sedimentary basin; vertical exaggeration ca. 2×; modified after [14]. (**c**,**d**) Stratigraphic log of cores SI3 and SV, and position of sediment samples. (**e**) Correlation among stratigraphic and hydrogeological units.

*2.2. Subsurface Stratigraphy of the Como Urban Area*

The subsoil of Como is composed of unconsolidated late-glacial and Holocene sediments representing various depositional environments [14]. The sedimentary sequence, starting from the top, is composed of (Figure 2):

- 1–10 m of heterogeneous reworked material with archeological remains (Unit 1); in the lake-shore area, a silty and highly compressible sub-unit (1b) has been recognized within the anthropogenic sediments.

- Alluvial sands and gravels (Unit 2) are present down to a depth of 15–24 m.
- up to 30 m of palustrine organic and highly compressible silts occur (Unit 3), which have been dated between ca 4 and 18.5 ka BP at Piazza Verdi site.
- Below 40–60 m depth, distal (silty) glaciolacustrine sediments with dropstones are present (Unit 4) that overlie coarser proximal deposits (Unit 5).

*2.3. Hydrological and Hydrogeological Setting*

The regional flow direction of both surface water and groundwater is from north to south, i.e., from the Alps toward the Po Plain. The Adda river is the main tributary of Lake Como and enters the lake at its northern tip; the Adda River continues its course at the south end of the Lecco branch of the lake (Figure 1a). Since the Holocene, the Como branch has had no outlet [15,30]. Thus, the area of the modern city of Como, proximal to the lake, represents a good sedimentary trap favoring the deposition of lacustrine and, later on, alluvial deposits. On a local scale, the water in the Como urban area flows from south to north, draining toward the local base level represented by Lake Como. Annual rainfall in the Como area is about 1500 mm/yr (average taken from 1891–1900 [31]); this rainfall regime favors the recharge and circulation of groundwater mainly in the carbonate sequence, which is highly permeable due to both fracturing and karstification [23]. The Como plain is drained by two streams, Valduce creek and Cosia stream (Figure 1b), which are presently artificially buried, and have an underground flow path in the urban area down to the mouth. Valduce creek provides an ephemeral flow and is mainly recharged by a small karst basin, whereas Cosia stream drains a larger watershed area (about 33.5 km$^2$) with measured flows up to 33 m$^3$/s during its maximum flow regime [32].

Groundwater circulation in the urban subsurface area is restricted to two different aquifers: a shallow phreatic aquifer and a deeper confined one. The shallow phreatic aquifer (up to 25 m thick) is hosted in Units 1 and 2 (Figure 2e); the presence of the low-permeability sub-unit 1b was ascertained along the lakeshore and, due to its limited lateral continuity, does not influence groundwater circulation at the basin scale. The water table in the shallow aquifer, strictly correlated with precipitation and changes in the hydrometric level of the lake, progressively decreases from south to north. Transmissivity of this aquifer was calculated as 2–5·10$^{-3}$ m$^2$/s and specific flow rate was measured as 0.9–6.5 L/s·m$^2$ [32]. Differently, the recharge areas and the main hydrochemistry of this aquifer are still unknown and were not investigated in previous studies.

The deeper confined aquifer, subdivided into several lenses or layers due to lateral heterogeneity, is located in the proximal sandy glaciolacustrine deposits (Unit 5) and is confined by the overlying sequence of clays and silts (Units 3 and 4 in Figure 2) [14,33]. Artesian wells were extensively exploited for residential and industrial use until 1980, when aquifer exploitation was prohibited due to the damage in the urban area caused by subsidence. Currently, this aquifer present an overpressure of hydraulic head [15]. The transmissivity and specific flow rate of the deeper aquifer were measured as 1·10$^{-3}$ m$^2$/s and 1.5–7.5 L/s·m$^2$, respectively [32]. However, the recharge areas of this aquifer were never investigated in detail. As a result, a conceptual model of the deep aquifer was not developed and the contributions of the possible water sources (i.e., meteoric water, the lake and the karst areas surrounding the plain) were not quantified in previous studies [32].

## 3. Materials and Methods

*3.1. Water Sampling and Chemical Analyses*

Water samples were collected in 12 localities spanning a range of surface and groundwater sources (Figure 1b). Sampling localities include water from Cosia creek sampled upstream from the urban area (samples CC1 and CC2) and just after entering in the urban area (CC3); lake surface water, collected at the outlet of Valduce creek (sample

L1); water from the shallow phreatic aquifer (samples S1–S7); and water from the deeper confined aquifer (samples D1 and D2).

Water samples from Cosia creek and from the aquifers were collected in spring (May) and fall (October and November) 2015. Two surface water samples of the lake were then collected in October 2020 and May 2021. Overall, a total of 37 water samples were collected and analyzed. Table S1 in Supplementary Material provides more details on the sampling dates and coordinates for each of the locations.

Samples from the shallow unconfined aquifer were collected after well purging, performed by pumping a volume equivalent to three times the volume of water found in the well during sampling. After purging, samples were collected by pumping with a 12 V bladder pump [34]. For water collected from the confined aquifer, samples were obtained after purging using the above process, then collected from the overpressured flow of water in the well case.

Physicochemical parameters (pH, temperature, and electrical conductivity) were measured directly on-site using probes: pH and temperature were measured with a HANNA Instruments HI 9025 pH meter, whereas electrical conductivity was measured with a HANNA Instruments HI 9033 conductivity meter. Water samples were filtered directly on site and collected in low density polyethylene (LDPE) bottles pre-cleaned with NALGENE detergent, whereas samples for cations were collected in LPDE bottles pre-cleaned with a 2% $HNO_3$ solution and acidified directly on site adding 100 μL of 0.1 M $HNO_3$. All bottles were kept away from light and refrigerated until laboratory analyses were conducted. Once in laboratory, samples for trace element analysis were acidified using distilled ultra-pure $HNO_3$ [35].

The laboratory water analyses include:

- Carbonates and bicarbonates through titration with 0.01 M HCl;
- Other major anions and cations ($F^-$, $Cl^-$, $NO_3^-$, $SO_4^{2-}$, $Na^+$, $K^+$, $Mg^{2+}$, and $Ca^{2+}$) with ion chromatography (metrohm EcoIC), with all ionic balances below the ±10% range;
- $SiO_2$ through the spectroscopic method ASTM D859-16, using a Merck Spectroquant Nova UV-visible spectrometer [36];
- PTEs (Li, V, Mn, Co, Ni, As, Rb, Sr, Ag, Cd, Pb, and U) with ICP-MS (Icap-Q ICP-MS, Thermo Fisher Scientific, USA). All samples were spiked with 1 μg/L of Rh as internal standard;

$\delta^{18}O_{H2O}$ using the $CO_2$-$H_2O$ equilibration method, and online chromium reduction using continuous flow isotope ratio mass spectrometry for $\delta^2H_{H2O}$. Results for both $\delta^2H$ and $\delta^{18}O$ $H_2O$ are reported in the usual delta notation against the Vienna Standard Mean Ocean Water (V-SMOW). The analytical error of these measurements was ±0.1‰ and ±1.0 ‰ for $\delta^{18}O_{H2O}$ and $\delta^2H_{H2O}$, respectively [37].

All samples from major ions, $SiO_2$ and trace elements were analyzed with three sets of replicates ensuring a relative standard deviation beneath ±5%. More details about chemical methods for water sampling and analyses, as well as QA/QC protocols, can be found elsewhere [38,39].

*3.2. Sediment Sample Collection and Analysis*

Six fine-grained sediment samples were collected from 2 cores drilled in the study area. Four samples were taken from core SI3, drilled on the lakeshore (Figure 2a). Samples were collected at depths ranging between 12 and 24 m below surface, from Units 1, 1b and 3 (Figure 2c). Two samples were collected from core SV1 drilled in Piazza Verdi, at depths of 56 and 65 m, representative of Unit 5 of the deep aquifer (Figure 2d). Gravels and impurities were directly removed by hand, then samples were air-dried in oven at 80 °C, ground using an agate ball mill and sieved using a 0.5 mm mesh nylon sieve. The sediments samples were then labeled as in Figure 2c and d, where "z" indicates the core depth core from which the samples were collected.

To analyze the elemental content in the solid samples, they were first acid-digested using a mixture of HNO₃ and HF, in order to dissolve also the silicate-bond metals. About 100 mg of dried sample was weighed and placed in a Teflon vessel, 3 mL of HF and 2 mL of HNO₃ were added, and the solution was heated for 20 min at 180 °C. Afterwards, dissolved samples were diluted with ultrapure water and stored in a refrigerator at 4 °C. Then, dissolved samples were analyzed (as for water samples) using a Thermo-Fisher scientific Icap Q ICP-MS for Li, V, Mn, Co, Ni, As, Rb, Sr, and U, adding 1 µg/L of Rh as internal standard. All dissolved sediment samples were analyzed with three sets of replicates ensuring a relative standard deviation below ±5%.

Powdered samples were also analyzed for their mineral composition using a Siemens D5000 X-ray diffractometer (XRD), equipped with a Cu kα source at 40 kV, 40 mA. Samples were analyzed with a 2 mm slit in the 5–60° 2θ range, with a step of 0.02° and a scan rate of 1.2°/min. Data were then processed with the Xpert highscore software to analyze the main crystalline phases and their relative abundance.

### 3.3. Data Analysis

#### 3.3.1. Principal Component Analysis (PCA)

Initial data analysis of the water chemistry (i.e., physicochemical parameters, major ions, and trace elements) was conducted using PCA, a multivariate statistical technique that generates new uncorrelated variables named principal components starting from a linear combination of the analyzed variables. This technique is useful in showing the range of variance within the dataset and widely applied in environmental sciences [2,40]. Afterwards, original variables (loadings) and samples (scores) can be plotted in the generated principal component space, to illustrate correlation among them [38,41,42]. These features make PCA a valid tool to interpret complex environmental systems, such as hydrogeological basins [1,6]. Prior to running the PCA, all values beneath detection limits were substituted with LOD/10 values to obtain a numerical value representing the uncertainty of below-LOD values [43,44], then data were normalized through z-score following Equation (1):

$$z = \frac{x - m}{s}$$

(1)

where $z$ is the normalized value, $x$ is the original measured value, $m$ is the mean value and $s$ is the standard deviation of the measured variable in all samples. This normalization helps to highlight changes in variables among different collected samples, enhancing a normal distribution of the dataset and highlighting possible geochemical anomalies [42,45].

#### 3.3.2. Water-Sediment Partition Coefficient

A partition coefficient $K_r$ was calculated between PTEs concentration in water and sediment [39,46,47] expressed as in Equation (2):

$$K_r = \frac{C_s}{C_w}$$

(2)

where $C_s$ indicates the elemental concentration value in solid phase (in mg/kg) and $C_w$ indicates the elemental concentration in water (in µg/L). More specifically, in this paper we calculated the $K_r$ coefficient using the PTE concentration for every water sample of the deep and the shallow aquifer as $C_w$, and the average concentration in solid samples from acid digestion in the various sediments hosting the aquifer as $C_s$ (obtained from samples SED1-Z12, SED1-Z14, SED1-Z17, and SED1-Z24 for the shallow aquifer and samples SED2-Z56 and SED2-Z65 for the deep aquifer, respectively). Through this ratio, we could quantify how likely the elemental concentration observed in water reflects the available concentration in sediment, or whether another source is involved in the concentration

observed in water (i.e., atmospheric deposition, mixing with water from different aquifers).

## 4. Results and Discussion

### 4.1. Water Chemistry

The values of the main physicochemical parameters and major dissolved ions clearly distinguish among the samples collected in the Cosia creek, the shallow and the deep aquifer. Samples from the deep aquifer show (i) a lower variance in both temperature and conductivity through the different sampling periods (i.e., indicating a slower response to climatic variables) and (ii) a lower value of conductivity than both the shallow unconfined aquifer and surface water samples (Figure 3a,b). The detailed results of different analyses performed for each water sample collected are available as Supplementary Material, in Tables S2 and S3.

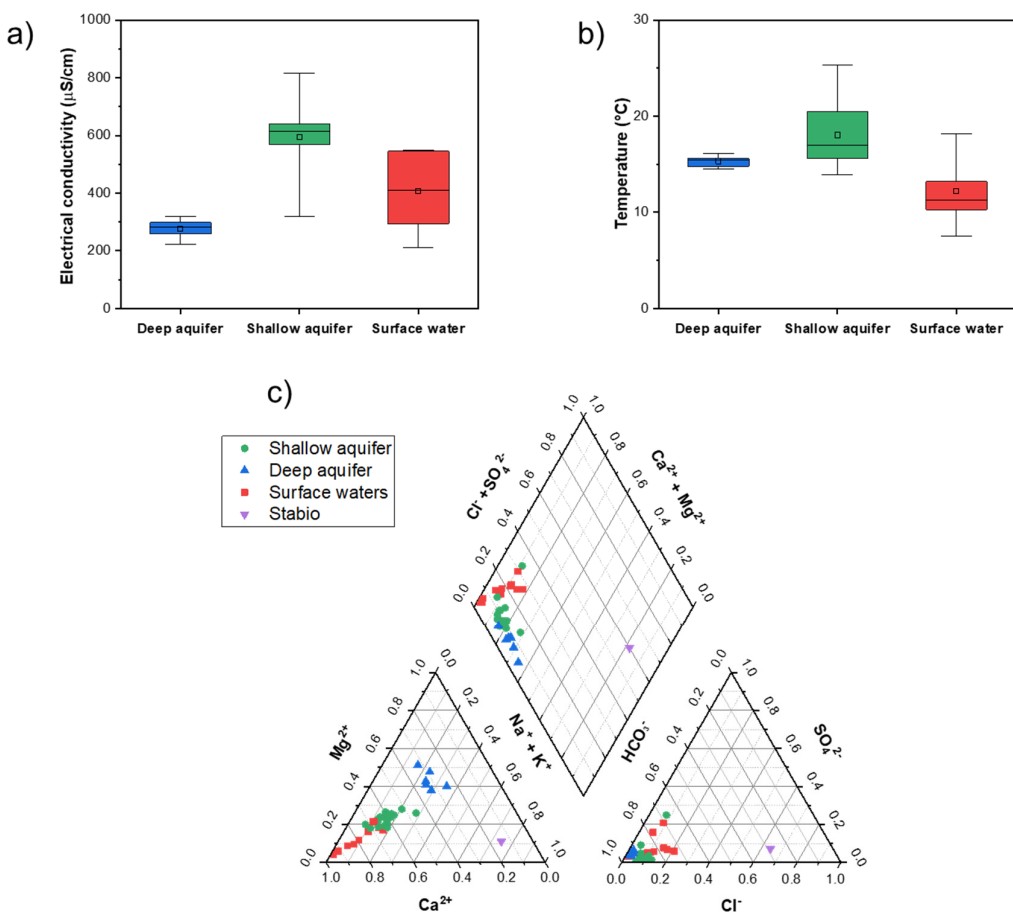

**Figure 3.** Box and whisker plot of electrical conductivity (**a**) and temperature (**b**) of the different aquifer and surface water samples. Boxes indicate the 25th–75th percentile range, whiskers indicate the minimum–maximum range, squares indicate the mean value and horizontal lines the median. (**c**) Piper plot of all the collected samples.

The Piper plot shows that all water samples present a dominance of $Mg^{2+}$-$Ca^{2+}$ and carbonate (Figure 3c), as expected considering that carbonate lithologies dominate in the study area and, in particular, in the Cosia creek watershed. However, the cation plot clearly indicates an enrichment in $Na^+$, $K^+$, and $Mg^{2+}$ with a decreasing trend of $Ca^{2+}$ moving from the surface water toward the shallow and the deep aquifers. The deep aquifer samples do not form a continuum with the other samples, but instead plots

separately from other samples along this trajectory. The enrichment in $Na^+$, $K^+$, and $Mg^{2+}$ could be due to either a longer residence time in the deep sediment, or a recharge area that is less dominated by carbonate bedrock, in contrast with the karstically-derived shallow aquifer. Potentially, the main recharge area for the deeper aquifer could either be the south-west area of the Como basin, which consists of Gonfolite Group composed of metamorphic and plutonic rocks [27], or the south-east part of Como basin, where glacial deposits are widespread [23].

Moreover, as a comparison, previously published major ion data for the Stabio hydrothermal water are reported in the Piper plot in Figure 3c [26]. Despite being located only 10 km far from Como, along the Gonfolite Backthrust, the Stabio waters show a completely different composition with respect to all the Como samples. This suggests a different water source of the thermal Stabio water, or at least a partial mixing of these waters with other fluids in the aquifers of the Como area.

The multivariate approach using PCA with physicochemical parameters, major ions and trace elements as input variables helped delineate the main geochemical properties and trends of the aquifers and surface waters. Most of the major ions and some trace elements show highly positive values of PCA component 1, which explains 38% of the total variance (Figure 4a), whereas some trace elements (namely As, Li, V, and U) show negative values of PCA component 1. Moreover, PCA component 2, which explains 18% of the total variance, shows that As, Li, Sr, and $F^-$ covary and separate well from other dissolved elements. The score plot (Figure 4b) indicates that the samples with the highest abundance As, Li, Sr, and $F^-$ (i.e., elements with high positive loadings of principal component 2) are from the deep aquifer samples, which is a geochemical attribute that distinguishes them well from other samples. In contrast, samples collected from both the surface water (Cosia Creek and the lake Como) and the shallow aquifer score with low or slightly negative values of PCA component 2. Instead, samples of the Cosia creek and the lake are distinguished by a higher load of $NO_3^-$ and $SO_4^{2-}$ (possibly related with a higher load of nutrients). Trace elements therefore provide a means to further distinguish shallow and deep aquifers.

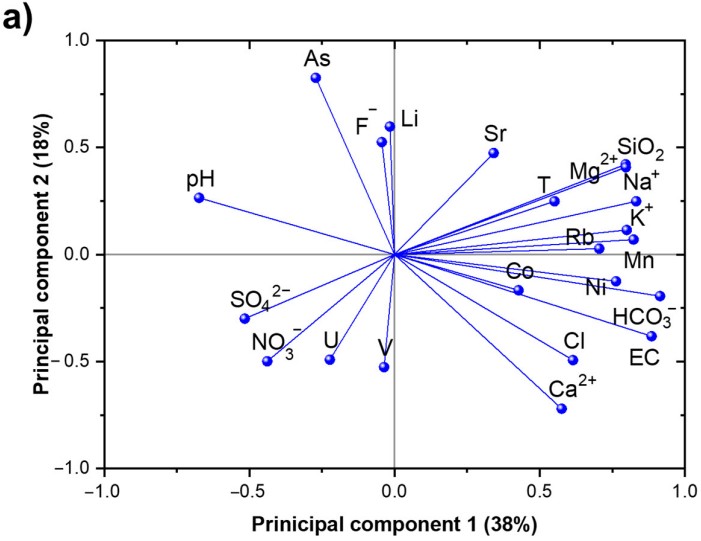

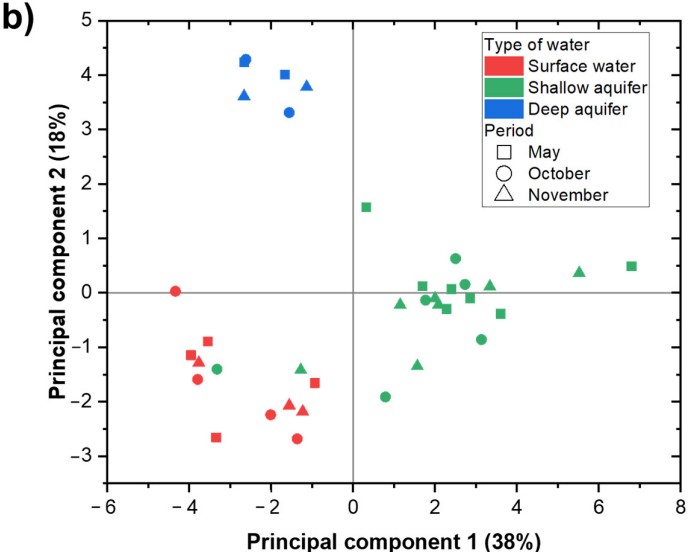

**Figure 4.** (**a**) PCA variable loadings and (**b**) score plot of principal component 1 (explaining 38% of the total variance) vs. component 2 (explaining 18% of the total variance) for all the analyzed sites.

The water collected in the deep aquifer shows a higher content of As, Sr and Li, while containing a lower value of Mn and Ni. The co-occurrence of these elements (especially As) can be indicative of the weathering of various crystalline components present in the sediments of the study area, since the sediment hosting the deep aquifer (Unit 5) is mainly derived by glacial deposition of crystalline rocks from the Alps [39,48,49]. However, the high load of elements such as As and Li can be possibly related with the interaction of deep seated hydrothermal fluids, since these elements are often related with volcanic and thermal aquifers [10,50]. Therefore, the geochemical characterization of sediment samples is discussed to help further distinguish the possible source of these elements in the two aquifers, and is discussed in the following section.

Finally, while the score plot (Figure 4b) shows a strong distinction between the aquifers, it does not show a specific seasonal influence. Water collected in different months are scattered, while samples collected from the three different water types (i.e.,

the two aquifers and surface water) are generally well separated. This lack of seasonal influence on the aquifer hydrochemistry further supports the existence of different pathways for water recharge for the shallow and deep aquifers, as already pointed out by major ions composition (Figure 3c). Moreover, Ag, Cd, and Pb showed concentrations below the detection limits in more than 50% of the analyzed water samples (Table S3 in Supplementary Material) and were therefore excluded by the PCA data treatment. This issue indicates a general low level of pollution in the Como aquifers [39,51].

### 4.2. Sediment Chemistry and Mineralogy

Table 1 shows the mineralogical composition of six samples obtained through XRD, while the Supplementary Material presents data on trace elements analyzed through acid digestion and ICP-MS (Table S4). Four samples were retrieved from the core SI3 (Figure 2): the sample at 12 m depth belongs to the anthropogenic fill of Unit 1, hosting the shallow water circulation; the sample at 14 m depth belongs to sub-unit 1b, which is a lens of fine sediments within Unit 1. The samples at 17 and 24 m depth represent the fine sediments of Unit 3, which have a lower relative permeability. Another two samples were collected from the core SV1: one was taken at the base of Unit 4 (56 m depth) and the other within Unit 5 (65 m depth), which hosts the deep aquifer.

The sedimentological data consistently highlight a difference between the units hosting the shallow unconfined aquifer, the aquitard and the deep confined aquifer. Deeper samples belonging to Units 4 and 5 show a higher concentration of antigorite and chlorite. This fact is explained by the provenance of sediment in Units 4 and 5, i.e., glaciofluvial deposits deriving from the erosion of lithologies of the Central Alps; more specifically, the antigorite indicates a source from the Pennidic Domain. The trend in trace elements (Table S4 in Supplementary Material) shows an increase in some elements (e.g., Li, Mn, Ni, and As) with depth. This result is in line with data on the main mineralogy: the increase in Li concentration is related to the more abundant feldspars and muscovite [52,53], whereas the increase in Ni, Mn, and As can be justified by the high concentration of mafic rock clasts in the sediment [47,54]. The presence of mafic rock clasts in the deep sediments is indeed marked by antigorite and K-feldspars (not detected in the shallower units) and by the increase in albite, chlorite, and muscovite.

Magnetic susceptibility and bulk density were also measured on core SV1 with a resolution of 1 cm [55]. The two parameters are covariant and suggest that magnetic susceptibility mainly reflects events of increased clastic sedimentation. Some peaks in magnetic susceptibility at the top of Unit 4 are interpreted as dropstone events, while high values in Unit 2 is indicative of a strong detrital fraction, as expected in alluvial sediments [55].

**Table 1.** Relative abundance of the main minerals in the sediment cores at different depths, obtained from XRD analyses performed on 4 samples from core SI3 and 2 samples from core SV1 (* symbols indicate the relative abundance, ranging from * indicating scarce amount to **** indicating a major abundance). Sedimentological units of all samples are also reported. See Figure 2 for core location and stratigraphy of the sampled cores. Refer to Figure S1 in Supplementary Material for XRD spectra.

| Sample | SED1-Z12 | SED1-Z14 | SED1-Z17 | SED1-Z24 | SED2-Z56 | SED2-Z65 |
|---|---|---|---|---|---|---|
| Unit | 1 | 1b | 3 | 3 | 4 | 5 |
| Quartz | **** | *** | *** | *** | *** | ** |
| Muscovite | * | ** | * | ** | ** | ** |
| Albite | * | * | * | ** | *** | ** |
| Anorthite | * | * | ** | ** | – | – |
| K-Feldspars | – | – | – | * | * | * |
| Amphibole | traces | – | – | traces | * | * |
| Calcite | *** | * | * | * | ** | * |

| | | | | | | |
|---|---|---|---|---|---|---|
| Chlorite | - | * | * | ** | ** | * |
| Antigorite | - | - | - | - | * | * |

### 4.3. Water Sediment Partition and Possible Mobility of Elements

The mobility of different trace elements in the groundwater of the Como aquifers can be analyzed in greater detail through use of $K_r$ values (Figure 5). Only a few differences are observable regarding the shallow and the deep aquifers. Two elements (Sr and As) show average negative log $K_r$ values, representing an enrichment derived from other sources than water–sediment interaction in the hosting substrate. With regard to As, the presence of negative $K_r$ values (especially in the deep aquifer) is related with two possible phenomena: (i) a dissolution from the bedrock in the recharge area of the aquifer and (ii) the possible mixing with thermal fluids uprising from deeper hosting rock. In contrast to As, Sr shows negative values both in the shallow and deep aquifers. We argue that the release of this element can be related to the widespread karst circulation in the carbonate lithologies diffusely present in the area. As a proof, when compared to other trace elements, the concentration of Sr is relatively high also in surface water samples (with an average value of 182 µg/L, see Table S3 in Supplementary Material).

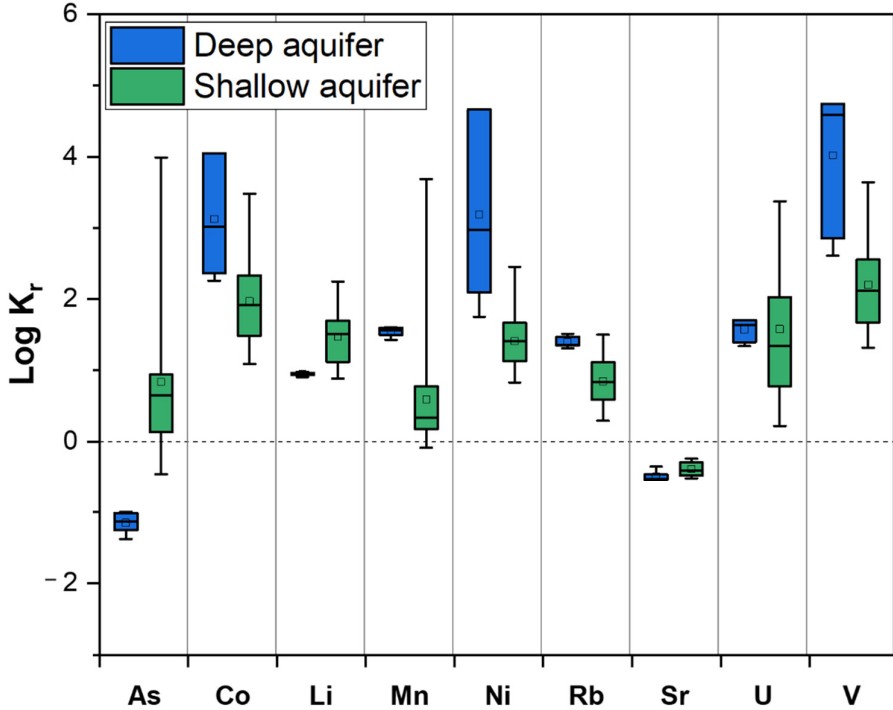

**Figure 5.** Range of $K_r$ values (in logarithmic scale) of the different trace elements analyzed in water samples of the shallow aquifer (indicated in green) and the deep aquifer (indicated in blue). Squares indicate the average value, horizontal lines the median value, boxes indicate the 25th–75th range, and whiskers indicate the minimum-maximum range.

Lithium, another element possibly enriched by hydrothermal waters, shows relatively high concentrations in the sediment of the study area, possibly derived by the dissolution of feldspars, quartz and biotite, known to release Li during weathering [52], which are abundant in the sediment of the Como area (Table 1). This can explain the observed concentration in water with higher values in the deep aquifer.

Vanadium, Co, Ni, and Mn show relatively high values, especially in the confined aquifer (Log $K_r$ > 2), indicating limited dissolution from the solid phase. As stated in the

previous paragraph, the presence of minerals such as antigorite and chlorite in the sediments hosting the deep aquifer justifies a relatively high load of these elements, which are usually not easily dissolvable and bonded to silicate phases, especially when considering the low oxidation potential in the confined aquifer [47,54,56,57]. In contrast with the deep aquifer, these elements show a higher mobility in the shallow aquifer (especially Mn). This can indicate that they are derived from the carbonate fraction of the sediment, which is more prone to dissolution [58]. The carbonate fraction seems to be particularly abundant in the first few meters of the sediment fill in the Como basin (Table 1).

### 4.4. Isotopic Analysis: Reconstruction of Recharge Areas

The $\delta^{18}O$ and $\delta^{2}H$ plot (Figure 6) highlights a good linear correlation for the samples, along the slope of the global meteoric line [59]. We can therefore identify meteoric water as the main source of recharge for the aquifers [60]. Most of the samples of the shallow aquifer cluster at lower values of $\delta^{18}O$ and $\delta^{2}H$, pointing to a recharge area with elevations located at a maximum of about 1000 m a.s.l. Only two samples (S3 and S4) showed a completely different composition, both from the October sampling campaign. This effect can be possibly driven by direct infiltration of rainfall into the shallow aquifer. Rainfall is particularly abundant in the fall period in the study area. The isotopic composition of Stabio waters (from [26]) are also plotted in Figure 6, showing that the recharge elevation of this thermal spring is similar to that of the shallow aquifer.

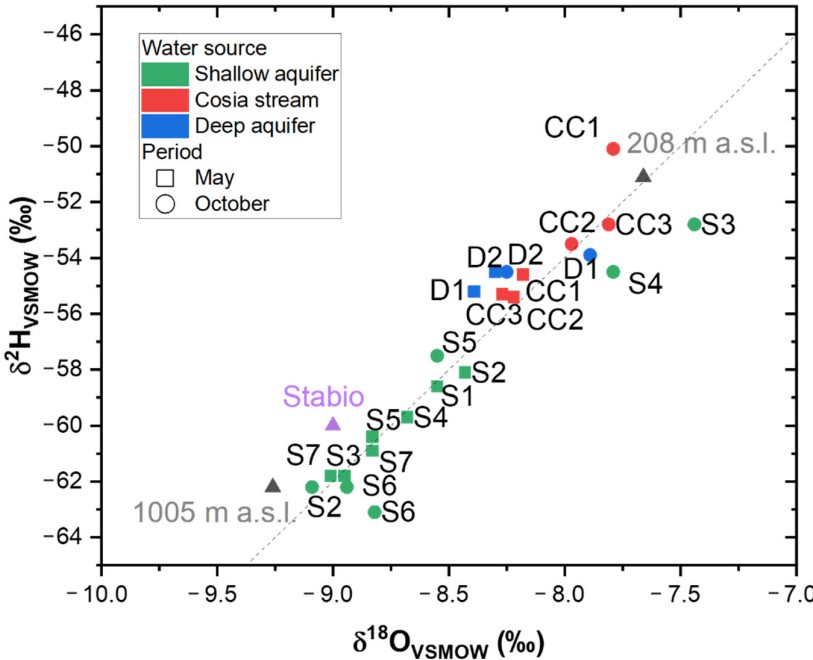

**Figure 6.** Isotope plot of the Cosia Creek, the shallow and the deep aquifer samples, sampled in May and October 2015. Shaded line indicates the equation of the Global Meteoric water line and 2 points of rain stations in Northern Italy (Andalo for 1005 m a.s.l. and Darfo Boario for 208 m a.s.l., respectively; [59]).

Interestingly, the estimated elevation of recharge areas of the deep aquifer samples is similar to those from Cosia creek, and of lower elevation than the shallow aquifer and Stabio waters. Additionally, the D1 site showed a higher variance in seasonal composition, in contrast from the other deep aquifer site D2. Given the similarities in recharge elevation

between the deep aquifer and Cosia stream, the deep aquifer in the Como basin is likely fed by infiltrations into the plain in the southern area of Como, which has this elevation range. The Cosia creek samples, moreover, show a seasonal trend: all the samples collected in spring period showed a depletion of heavier isotopes, while in fall period samples are mainly connected with lower recharge areas. This behavior indicates that runoff from snow melt contribute to the recharge of this creek in the spring [61], mixing with a lower altitude aquifer feeding the creek (highlighted by the enrichment of autumn samples).

Lake water samples were not analyzed regarding their water isotopes, because of the wide and mixed number of springs feeding Lake Como [62] (as for other big lakes in the southern Alps, e.g., [63]) and thus playing as a confounding factor in the interpretation of lake water isotopes.

### 4.5. Conceptual Model of the Como Aquifers: Possible Recharge Areas and Source of Elements

Accounting for the hydrochemical features of water samples presented in the previous paragraphs, it is possible to infer the main recharge areas and to develop the most plausible conceptual model of the Como basin sub-surface flow. The pathways of groundwater circulation depend on the difference in relative permeability among the host sediments, which, in turn, mainly results from the degree of fracturing of bedrock, and from the stacking of sedimentary units the stratigraphic juxtaposition of layers with different grain sizes. Additionally, structural elements such as the Gonfolite Backthrust influence groundwater circulation, and may have a role in either enhancing or hampering water flow.

Figure 7 illustrates our preferred conceptual model, highlighting recharge areas and the supply to the different sectors of the shallow and deep aquifers.

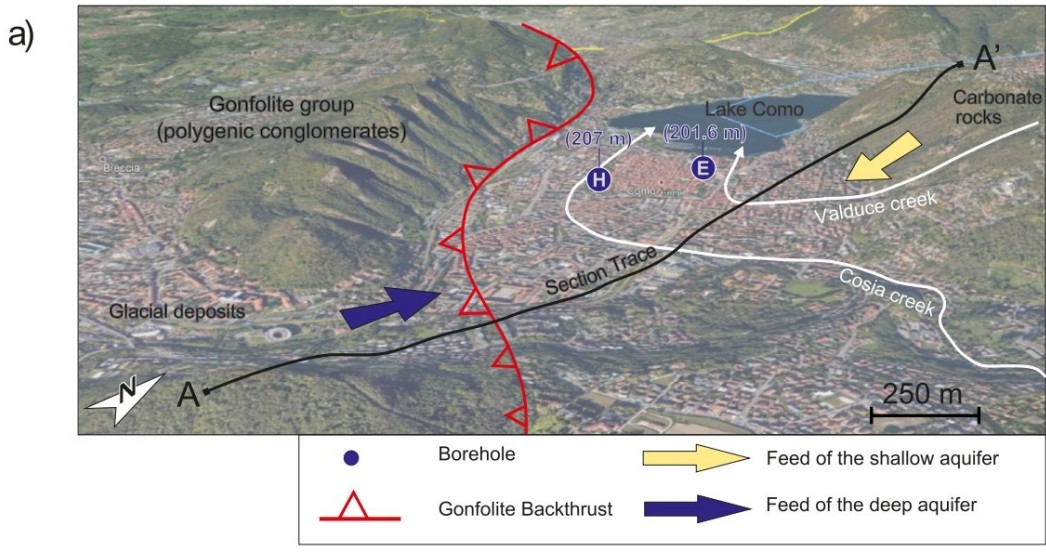

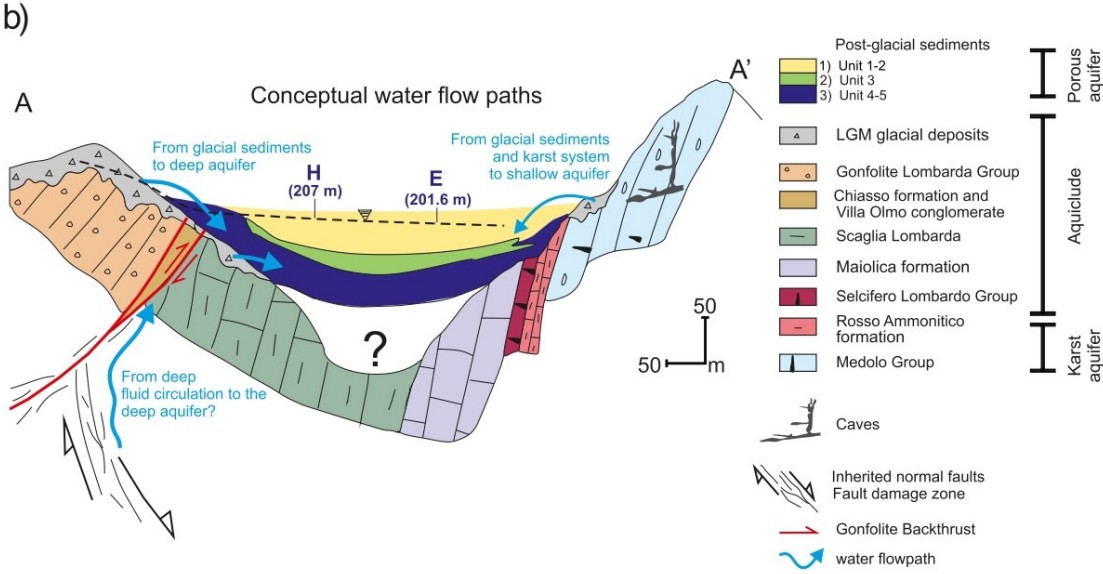

**Figure 7.** (**a**) Recharge areas and flow pathways (arrows) feeding the shallow and deep aquifers in the Como region. Section A-A' is indicated with a black line (see Figure 1b for its detailed spatial location). Boreholes H and E are shown (the reader is referred to Figure 2a for their location on map) together with the elevation of the hydraulic head of the deep aquifer (in parentheses, expressed in m a.s.l.). Basemap: Google Earth imagery. (**b**) Sketch of the conceptual model for water circulation and recharge areas through the section of Como basin along the section A-A'. Boreholes H and E are also indicated, including the current inferred line of hydraulic head (black dashed line). Details on the different geological formations can be found in [23].

Starting from the shallow unconfined aquifer, the main recharge elevations (ca. 1000 m a.s.l.) indicates that the most likely recharge areas are situated north-east of the city of Como. Moreover, the calcium carbonate main hydrological facies strongly supports a recharge from the north-east area, dominated by carbonate rocks. The chemical trend of trace elements is similar to Cosia creek and partly reflects the lake hydrochemistry,

indicating a fast flow from the sediment to the lake, as confirmed from hydraulic head mapping of the shallow aquifer [14]**.**

In contrast, the deep aquifer water samples are distinct from the shallow aquifer and surface water samples, especially with a marked increase in the relative abundance of the major element Mg, lower concentration values of Mn and Ni and higher values of As and Li. The weak seasonal difference suggests a slow flow rate and high residence time in the aquifer. Moreover, isotope data suggest that the recharge elevations of the deep aquifer samples are lower than the ones of the shallow aquifer, confirming a separate water source for this aquifer compared to the shallow one.

The deep aquifer was heavily exploited beginning the mid-1900s. Piezometric measurements in 1975–1977 attest that the piezometric surface measured at borehole E (Figure 7) was ca. 1 m below the topographic surface, while it was measured as 3–6 m below the surface at the piezometer in borehole H [64]. In 1980, groundwater exploitation was prohibited and since then the deep aquifer pressure has been restored sufficiently to return artesian flow. Measurements taken in 2014 using a pressure gauge have shown that the hydraulic head is at ca. 201.5 m a.s.l. at borehole E and at 207 m a.s.l. at borehole H, suggesting a flow direction toward the lake (Figure 7). Consequently, the most likely recharge areas of the deep aquifer are the Pleistocene glacial sediment surrounding the southern part of Como basin. Alternatively, the area hosting the Gonfolite group can possibly contribute in the recharge of the deep aquifer, but the low hydraulic conductivity of the unit suggests a relatively low contribution in the recharge of the deep aquifer, considering that this unit does not contribute to any springs in the study area [23]. Finally, while it is unlikely that there is a significant upwelling of thermal fluids in this setting (in contrast to the Stabio locality), it is not possible to completely rule out a partial injection of thermal fluids from the carbonate basement of the Como area (e.g., the Scaglia Lombarda units, Figure 7b). In fact, while the recharge area and the major ions of Como waters identified in these results are evidently distinct from Stabio ones (Figures 3 and 6), the concentration of some thermal related trace elements (especially As and Li) is higher in the deep aquifer than both the surface waters and the shallow aquifer, possibly justifying some influence of thermal fluids [42,65].

### 4.6. A Focus on As Geochemical Anomaly in the Deep Aquifer

The results showed a clear geochemical separation of the deep aquifer of Como basin with respect to the surface waters, which permit to discriminate a different recharge area for water residing in this aquifer. Among these differences, it is worth considering an anomalously high concentration of As. This element in the deep aquifer shows values higher than tenfold the World Health Organization (WHO) threshold limits for human use of 10 μg/L (with an average value in the deep aquifer of about 150 μg/L, Figure 8) and well above other spring waters in northern Italy [66], but cannot be solely linked to the host sediment; the As concentration in the host sediments (Table S4 in Supplementary Material) is in fact similar to other values of floodplain sediments in other settings of northern Italy [66]. Arsenic, moreover, is the only element (aside from Sr) showing negative $K_r$ values (Figure 5). The negative Sr values can be linked to the carbonate fraction of the sediment but the As cannot. While we cannot rule out some local As anomalies present in the glaciofluvial sediment but undetected in the analyzed samples, the observed data of sediment concentration suggest that source of this element in the deep aquifer is not directly released from the hosting sediments of Unit 5 and the aquitard of Unit 4 (see Figure 2).

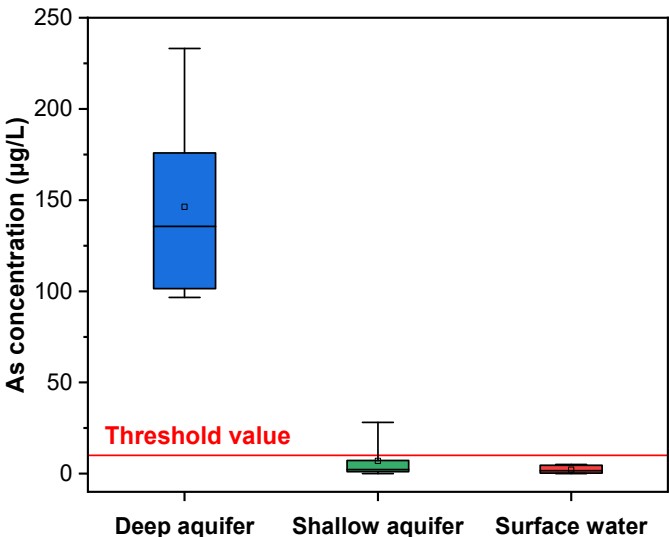

**Figure 8.** Box and whisker plot of As concentration values in the deep aquifer (in blue), the shallow aquifer (in green) and surface water (in red). Boxes indicate the 25th–75th percentile range, whiskers indicate the minimum–maximum range, squares indicate the mean value and horizontal lines the median. The red horizontal line indicates the threshold value for drinking water suggested by WHO.

Some potential sources of As external to the deep aquifer of Como basin may be considered, which include: (i) a possible anthropogenic release; (ii) the release from As-bearing minerals (mostly pyrite and arsenopyrite [67]); (iii) the release from alluvial and organic-rich deposits, which are known to contain As adsorbed onto Fe-hydroxides in settings close to our study area (i.e., the Po plain [66,68]), and (iv) the injection of deep seated thermal fluids [69].

Firstly, while a possible anthropogenic enrichment of this element can be potentially evaluated, the nature of the confined aquifer which shows a weak communication with the urbanized area and the low concentration of other possibly anthropogenically related PTEs (e.g., V, Cr, Ni, Cd, and Pb) makes this hypothesis unlikely.

Therefore, it is reasonable to consider the natural geological source of this element in the deep aquifer of Como. Given the available lithologic data for our study area (Tables 1 and S4) and the geological setting, the enrichment from dissolution of pyrite and arsenopyrite seems unlikely and can be ruled out. Additionally, the injection of deep thermal fluids also seems unlikely considering the differences with major ions and isotope with Stabio water. The relatively higher concentration of also Li and F⁻ in the deep aquifer (compared with the other water samples of the study area) can possibly justify an injection of a small amount of thermal water in the aquifer [70], with a consequent mixing with other freshwaters (as also observable from the PCA plot in Figure 4), but is not sufficient to explain the As anomaly. Considering the additional sources of As in settings close to our study area (e.g., [68,71]) the most likely mechanism is related to the desorption of this element from organic-rich deposits in a multilayered aquifer. The initial introduction of As can likely happen in the recharge area, where the reductive conditions of the confined aquifer highly favor its dissolution and consequent increase in concentration in the Como deep aquifer [72].

However, it is still not possible to infer a more specific mechanism of dissolution of this element. To confirm the most likely mechanisms of As dissolution and evaluate the

causes of this geochemical anomaly, more specific analyses (e.g., speciation in groundwater and in solid matrices [73]) and the comparison with other springs (for water) and cores (for sediments) in different localities close to the study area are needed. The presence of this toxic element warrants a careful and dedicated investigation, especially when considering the abundant exploitation of water for human use in the southern part of Como basin [74]. The data presented herein will possibly inspire a more detailed investigation of the setting of Como aquifer, in order to shed light on the causes of this As anomaly.

## 5. Conclusions

This study aimed to firstly characterize the groundwater composition of the multilayered urban aquifer underlying the city of Como, and secondly to validate the use of geochemical markers as tools for the reconstruction of recharge areas and conceptual models of subsurface flow. This study incorporates data from the hydrochemistry, flow, and sedimentary geochemistry, and frames the results in a regional geologic and tectonic context. The results obtained from water analyses highlight a clear difference between Como's shallow unconfined aquifer (characterized by a calcium-bicarbonate facies and a higher amount of Mn) and the underlying deep confined aquifer (characterized by lower conductivity, yet higher concentrations of Li, Sr, and As). These differences indicate different source waters and scarce communication between the two aquifers. The differences in source waters are further confirmed by a notable disparity in water isotopic composition, indicating distinct recharge areas. The deeper aquifer shares water isotopic values with the local Cosia stream, showing some seasonal change and a lower recharge elevation equivalent to the basin plain south of the city. In contrast, shallower aquifer shows a higher elevation recharge area. The analysis of sediments samples permits characterization of the main mineralogical and chemical features of the substrate hosting the aquifer, highlighting that most of the hydrochemical composition of the two aquifers can be justified by the geochemical mineralogical differences in aquifer host sediments, and confirming that local dissolution is the main source of elements in these aquifers.

From a water quality standpoint, the analyses highlighted high As concentrations in the deep aquifer. After considering the available data and the studies performed in similar settings, the main cause of the As anomaly is interpreted to be from the desorption of this element from organic-rich deposits. The effect of a possible injection of thermal waters can also be a factor, considering the tectonic setting, but cannot exclusively explain the high As concentration, given the distribution of other elements found in thermal waters nearby. More focused analyses are needed to validate the mechanisms associated with As dissolution, migration and enrichment.

**Supplementary Materials:** The following supporting information can be downloaded at: https://www.mdpi.com/article/10.3390/w14010124/s1, Figure S1: XRD spectra of sediment samples collected at different depths in core SED1 (a) and SED2 (b). Major peak patterns of the main minerals present in the samples are indicated by the abbreviations (Qz: quartz; Mu: muscovite; Al: albite; An: anorthite; K: K-feldspars; Amp: amphibole; Ca: calcite; Chlo: chlorite; Ant: antigorite), Table S1: Location and type of water sampled in different sampling points, Table S2: Physicochemical parameters and major ion data of all water samples, including ion balance, Table S3: Trace element concentration and isotopic values of all water samples, Table S4: Concentration of different metals in the sediment samples (average values of 3 replicates, relative standard deviation < 5%).

**Author Contributions:** Conceptualization, G.B., F.F. and S.T. (Silvia Terrana); formal analysis, F.F., R.G., D.S., and S.T. (Sara Trotta); data curation, F.A.L. and A.P.; writing—original draft preparation, G.B., F.F. and S.T. (Silvia Terrana); writing—review and editing, P.J.N., M.F.F., F.A.L., and S.T. (Sara Trotta); visualization, M.F.F., G.B. and D.S.; supervision, A.P. and A.M.M.; funding acquisition, A.P., S.T. (Silvia Terrana), and A.M.M. All authors have read and agreed to the published version of the manuscript.

**Funding:** This research received no external funding.

**Institutional Review Board Statement:** Not applicable.

**Informed Consent Statement:** Not applicable.

**Data Availability Statement:** Data is contained within the article or Supplementary Material.

**Acknowledgments:** The authors wish to thank the Como municipality for its support in sampling of groundwater and Simon Poulson for running isotopic analysis of water samples. Andrea Cattaneo, Marta Keller and Davide Campagnolo are also acknowledged for their support with XRD analyses.

**Conflicts of Interest:** The authors declare no conflict of interest.

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
