# Peer review of "Geochemical Markers as a Tool for the Characterization of a Multi-Layer Urban Aquifer: The Case Study of Como (Northern Italy)"

_water, doi:10.3390/w14010124_

Round 1
Reviewer 1 Report
Comments identified by line numbers
17 and 22 – Is “exploit(ation)” a proper word here?
25 – “37” – A sentence should not start with a number (it should be spelled out).
35 - “a tool for the recharge area” (definite article missing)
59, and elsewhere many times – I am not sure if the acronyms N, S, E, W and combinations are allowed without definition on their first appearance. At least, they should be used consistently. There are several cases where these were fully spelled out. Please, check the entire manuscript and be consistent.
76-77 - “have previously been found” (‘been’ missing)
81 - “understanding urban groundwater” (‘of’ redundant)
97 – Spell out ‘Fm.’ (Formation?). Is ‘Moltrasio Limestone Formation’ a correct official name? I have found only ‘Moltrasio Limestone’ or ‘Moltrasio Formation’ by a quick Google check.
131 - “red lines … indicate”
132 – A reference to the topo map should be included in References, if it is not there.
138 - “sub-unit (Unit 1b)” (‘;’ redundant)
142 - “18.5 ka BP” (‘ca’ redundant orin a wrong position)
154 - “1891-1900” – Is it correct? Are there no more present data available?
176-177 – The number should not be broken into two lines.
Figure 2 a) vs b) – C019 vs C19? – consistency!
199 - “Table S1 … provides”
221 – ASTM D859-16 – a reference for the method?
226-227 – More details on the isotopic analyses should be given (standards used, formulas used to calculate isotopic ratios, or suitable references).
228 – To what analyses does the given rsd relate? All analyses?
297, 300, 473 – When mentioning ions, charges should be given (or elements should be mentioned instead ions).
314 - “of electrical conductivity (a) and temperature (b) of the different” (change word order and do not capitalize ‘electrical’).
341 - “further distinguishing”
357 and 383 - “from the core” (def. article missing)
366 and elsewhere several times – I would not capitalize mineral names.
411 and 412 – ‘elements’ vs ‘ions’ (be consistent)
418 - “range, and whiskers” (comma missing)
425 and elsewhere several times - ‘asl’ vs ‘a.s.l.’ (be consistent)
427 - “further distinguishing”
429 - “plotted in Figure 6”
433 - “of the Global Meteoric”
465 - “areas are situated NE of the city”
466, 569 – Why is ‘facies’ italicized?
480 – I don’t understand well the 2nd half of the sentence. Is ‘at’ used or placed correctly?
483 - “gauge have shown”
488 - “conductivity of the unit”
489 - “contribute to any springs”
513 - “sediments (Table S4” (comma redundant)
518 - “as of yet” - ???
531 - “As adsorbed onto”
References – Check thoroughly the required format and especially journal abbreviations. I suspect that some may not be correct.
Author Response
We thank the reviewer for the efforts in reviewing our manuscript, detailed responses to the specific comments are listed as follows, joining specific comments requiring similar changes in text within a single response. We also kindly note that all the text lines indicated in the responses refer to the “marked change” version of the revised manuscript.
59, and elsewhere many times – I am not sure if the acronyms N, S, E, W and combinations are allowed without definition on their first appearance. At least, they should be used consistently. There are several cases where these were fully spelled out. Please, check the entire manuscript and be consistent.
We thank the reviewer for the comment. Accordingly, we used the extended spelling of cardinal points for the sake of consistency.
97 – Spell out ‘Fm.’ (Formation?). Is ‘Moltrasio Limestone Formation’ a correct official name? I have found only ‘Moltrasio Limestone’ or ‘Moltrasio Formation’ by a quick Google check.
We thank the reviewer for this comment. We now spell out the acronym Fm. As “formation” in lines 100-102.
The official name is “Calcare di Moltrasio” (Moltrasio Limestone). Moltrasio is a locality near Como where the formation is extensively outcropping; the formation is described in detail in Bernoulli, 1964 (D. Zur Geologie des Monte Generoso (Lombardische Alpen): ein Beitrag zur Kenntnis der südalpinen Sedimente Kümmerly & Frey, 1964). It was earlier referred to as “Lombardischer Kieselkalk” (Lombardy cherty limestone) by German-speaking authors; such name is now replaced by "Calcare di Moltrasio" in the official Italian geologic maps (Servizio Geologico d’Italia Carta Geologica d’Italia alla scala 1: 50.000 - Foglio 075, Como 2014).
132 – A reference to the topo map should be included in References, if it is not there.
221 – ASTM D859-16 – a reference for the method?
The following references were added for the topo map and for the Silica analytical method, respectively:
- Regione Lombardia Carta Tecnica Regionale Available online: https://www.geoportale.regione.lombardia.it/ (accessed on Jul 5, 2021).
- ASTM International Standard Test Method for Silica in Water. 2000.
226-227 – More details on the isotopic analyses should be given (standards used, formulas used to calculate isotopic ratios, or suitable references).
We thank the reviewer for this observation, we revised the text lines 241-244 and added a reference.
228 – To what analyses does the given rsd relate? All analyses?
We thank the reviewer for highlighting this issue, we revised the text as “All samples from major ions, SiO2 and trace elements were analyzed with three sets of replicates ensuring a relative standard deviation beneath ± 5%” in line 245 and added the detailed relative error of isotopic analysis in lines 243-244.
480 – I don’t understand well the 2nd half of the sentence. Is ‘at’ used or placed correctly?
We thank the reviewer for catching this. The sentence was rewritten as: “Piezometric measurements in 1975-1977 attest that the piezometric surface measured at borehole E (Figure 7) was ca. 1 m below the topographic surface, while it was measured as 3-6 m below the surface at the piezometer in borehole H” (lines 507-509).
17 and 22 – Is “exploit(ation)” a proper word here?
25 – “37” – A sentence should not start with a number (it should be spelled out).
35 - “a tool for the recharge area” (definite article missing)
76-77 - “have previously been found” (‘been’ missing)
81 - “understanding urban groundwater” (‘of’ redundant)
131 - “red lines … indicate”
138 - “sub-unit (Unit 1b)” (‘;’ redundant)
142 - “18.5 ka BP” (‘ca’ redundant or in a wrong position)
154 - “1891-1900” – Is it correct? Are there no more present data available?
176-177 – The number should not be broken into two lines.
Figure 2 a) vs b) – C019 vs C19? – consistency!
199 - “Table S1 … provides”
297, 300, 473 – When mentioning ions, charges should be given (or elements should be mentioned instead ions).
314 - “of electrical conductivity (a) and temperature (b) of the different” (change word order and do not capitalize ‘electrical’).
341 - “further distinguishing”
357 and 383 - “from the core” (def. article missing)
366 and elsewhere several times – I would not capitalize mineral names.
411 and 412 – ‘elements’ vs ‘ions’ (be consistent)
418 - “range, and whiskers” (comma missing)
425 and elsewhere several times - ‘asl’ vs ‘a.s.l.’ (be consistent)
427 - “further distinguishing”
429 - “plotted in Figure 6”
433 - “of the Global Meteoric”
465 - “areas are situated NE of the city”
466, 569 – Why is ‘facies’ italicized?
483 - “gauge have shown”
488 - “conductivity of the unit”
489 - “contribute to any springs”
513 - “sediments (Table S4” (comma redundant)
518 - “as of yet” - ???
531 - “As adsorbed onto”
We thank the reviewer for catching these misspellings and notify them. We carefully revised the text according to the reviewer’s suggestions.
References – Check thoroughly the required format and especially journal abbreviations. I suspect that some may not be correct.
We thank the reviewer for this comment, we checked and revised the reference list accordingly.
Reviewer 2 Report
The article is undoubtedly interesting and relevant; contains new hydrogeochemical data for a complexly constructed aquifer in the urbanized territory of the Como region. Using traditional hydrogeochemical methods and the method of principal component analysis, a significant difference in the chemical composition of groundwater of the shallow and deep aquifers is shown. A conceptual model of the chemical composition of groundwater in the region is considered, the main role in which is played by the difference in the recharge areas of the shallow and deep aquifers. The authors also argue that the hydrogeochemical data used can serve as markers to characterize main factors affecting natural water quality, as well as a tool for recharge area reconstruction.
At the same time, the negligence in the design of the article materials gives rise to doubt in the constructions of the authors. The authors are invited to perform a more in-depth processing of hydrogeochemical materials and the text of the article. I hope that the comments below will help this.
- Firstly, the issue of anthropogenic pollution of groundwater remained outside the attention of the authors. The role of the anthropogenic load of the surface and ground waters under consideration has not been assessed. This is especially true of the shallow aquifer, for which there is an increased electrical conductivity and temperature (Fig. 3 a, b).
Without considering anthropogenic factor, the authors' constructions regarding the exclusive role of natural factors in explaining the peculiarities of the chemical composition of the shallow and deep aquifers are not convincing.
There is no data on contaminating components, except for NO3. This is a significant disadvantage of the work.
- Fig. 1 and Fig. 2a) should be edited to clearly indicate the spatial position and the location of territories in Fig. 1a, 1b and 2a as well as Fig. 7. As presented, it is not clear where the coring wells are in relation to the water sampling locations? Where the section A - A' (Fig. 7) is located in relation to Fig.1b?
All geographical objects mentioned in the text (rivers, streams, valleys, etc) should be shown in these figures.
- L257_L283 (3.3.1. Principal component analysis (PCA)): A more detailed explanation of the need to use principal component analysis in this study is needed. Links to publications 33,35,36 are not enough, explanations are needed for the readers of the article.
- L270_275: the need to give a more detailed explanation of the method is obvious, since references to works [34,40,41] are not enough.
- L306: a short description of the Stabia thermal waters (temperature, discharge volume, chemical composition of water and gas) should be given. Link to publication 42 is not enough for the readers of the article.
- The material on hydrogeological conditions is insufficiently systematized. A description of the hydrogeological conditions is provided in the excerpts in Sections 2.3 and 4.5. A more complete and detailed description of the hydrogeological conditions of the area should have been provided.
The manuscript should be finalized for publication in the journal.
The article is undoubtedly interesting and relevant; contains new hydrogeochemical data for a complexly constructed aquifer in the urbanized territory of the Como region. Using traditional hydrogeochemical methods and the method of principal component analysis, a significant difference in the chemical composition of groundwater of the shallow and deep aquifers is shown. A conceptual model of the chemical composition of groundwater in the region is considered, the main role in which is played by the difference in the recharge areas of the shallow and deep aquifers. The authors also argue that the hydrogeochemical data used can serve as markers to characterize main factors affecting natural water quality, as well as a tool for recharge area reconstruction.
At the same time, the negligence in the design of the article materials gives rise to doubt in the constructions of the authors. The authors are invited to perform a more in-depth processing of hydrogeochemical materials and the text of the article. I hope that the comments below will help this.
- Firstly, the issue of anthropogenic pollution of groundwater remained outside the attention of the authors. The role of the anthropogenic load of the surface and ground waters under consideration has not been assessed. This is especially true of the shallow aquifer, for which there is an increased electrical conductivity and temperature (Fig. 3 a, b).
Without considering anthropogenic factor, the authors' constructions regarding the exclusive role of natural factors in explaining the peculiarities of the chemical composition of the shallow and deep aquifers are not convincing.
There is no data on contaminating components, except for NO3. This is a significant disadvantage of the work.
- Fig. 1 and Fig. 2a) should be edited to clearly indicate the spatial position and the location of territories in Fig. 1a, 1b and 2a as well as Fig. 7. As presented, it is not clear where the coring wells are in relation to the water sampling locations? Where the section A - A' (Fig. 7) is located in relation to Fig.1b?
All geographical objects mentioned in the text (rivers, streams, valleys, etc) should be shown in these figures.
- L257_L283 (3.3.1. Principal component analysis (PCA)): A more detailed explanation of the need to use principal component analysis in this study is needed. Links to publications 33,35,36 are not enough, explanations are needed for the readers of the article.
- L270_275: the need to give a more detailed explanation of the method is obvious, since references to works [34,40,41] are not enough.
- L306: a short description of the Stabia thermal waters (temperature, discharge volume, chemical composition of water and gas) should be given. Link to publication 42 is not enough for the readers of the article.
- The material on hydrogeological conditions is insufficiently systematized. A description of the hydrogeological conditions is provided in the excerpts in Sections 2.3 and 4.5. A more complete and detailed description of the hydrogeological conditions of the area should have been provided.
The manuscript should be finalized for publication in the journal.
The article is undoubtedly interesting and relevant; contains new hydrogeochemical data for a complexly constructed aquifer in the urbanized territory of the Como region. Using traditional hydrogeochemical methods and the method of principal component analysis, a significant difference in the chemical composition of groundwater of the shallow and deep aquifers is shown. A conceptual model of the chemical composition of groundwater in the region is considered, the main role in which is played by the difference in the recharge areas of the shallow and deep aquifers. The authors also argue that the hydrogeochemical data used can serve as markers to characterize main factors affecting natural water quality, as well as a tool for recharge area reconstruction.
At the same time, the negligence in the design of the article materials gives rise to doubt in the constructions of the authors. The authors are invited to perform a more in-depth processing of hydrogeochemical materials and the text of the article. I hope that the comments below will help this.
- Firstly, the issue of anthropogenic pollution of groundwater remained outside the attention of the authors. The role of the anthropogenic load of the surface and ground waters under consideration has not been assessed. This is especially true of the shallow aquifer, for which there is an increased electrical conductivity and temperature (Fig. 3 a, b).
Without considering anthropogenic factor, the authors' constructions regarding the exclusive role of natural factors in explaining the peculiarities of the chemical composition of the shallow and deep aquifers are not convincing.
There is no data on contaminating components, except for NO3. This is a significant disadvantage of the work.
- Fig. 1 and Fig. 2a) should be edited to clearly indicate the spatial position and the location of territories in Fig. 1a, 1b and 2a as well as Fig. 7. As presented, it is not clear where the coring wells are in relation to the water sampling locations? Where the section A - A' (Fig. 7) is located in relation to Fig.1b?
All geographical objects mentioned in the text (rivers, streams, valleys, etc) should be shown in these figures.
- L257_L283 (3.3.1. Principal component analysis (PCA)): A more detailed explanation of the need to use principal component analysis in this study is needed. Links to publications 33,35,36 are not enough, explanations are needed for the readers of the article.
- L270_275: the need to give a more detailed explanation of the method is obvious, since references to works [34,40,41] are not enough.
- L306: a short description of the Stabia thermal waters (temperature, discharge volume, chemical composition of water and gas) should be given. Link to publication 42 is not enough for the readers of the article.
- The material on hydrogeological conditions is insufficiently systematized. A description of the hydrogeological conditions is provided in the excerpts in Sections 2.3 and 4.5. A more complete and detailed description of the hydrogeological conditions of the area should have been provided.
The manuscript should be finalized for publication in the journal.
The article is undoubtedly interesting and relevant; contains new hydrogeochemical data for a complexly constructed aquifer in the urbanized territory of the Como region. Using traditional hydrogeochemical methods and the method of principal component analysis, a significant difference in the chemical composition of groundwater of the shallow and deep aquifers is shown. A conceptual model of the chemical composition of groundwater in the region is considered, the main role in which is played by the difference in the recharge areas of the shallow and deep aquifers. The authors also argue that the hydrogeochemical data used can serve as markers to characterize main factors affecting natural water quality, as well as a tool for recharge area reconstruction.
At the same time, the negligence in the design of the article materials gives rise to doubt in the constructions of the authors. The authors are invited to perform a more in-depth processing of hydrogeochemical materials and the text of the article. I hope that the comments below will help this.
- Firstly, the issue of anthropogenic pollution of groundwater remained outside the attention of the authors. The role of the anthropogenic load of the surface and ground waters under consideration has not been assessed. This is especially true of the shallow aquifer, for which there is an increased electrical conductivity and temperature (Fig. 3 a, b).
Without considering anthropogenic factor, the authors' constructions regarding the exclusive role of natural factors in explaining the peculiarities of the chemical composition of the shallow and deep aquifers are not convincing.
There is no data on contaminating components, except for NO3. This is a significant disadvantage of the work.
- Fig. 1 and Fig. 2a) should be edited to clearly indicate the spatial position and the location of territories in Fig. 1a, 1b and 2a as well as Fig. 7. As presented, it is not clear where the coring wells are in relation to the water sampling locations? Where the section A - A' (Fig. 7) is located in relation to Fig.1b?
All geographical objects mentioned in the text (rivers, streams, valleys, etc) should be shown in these figures.
- L257_L283 (3.3.1. Principal component analysis (PCA)): A more detailed explanation of the need to use principal component analysis in this study is needed. Links to publications 33,35,36 are not enough, explanations are needed for the readers of the article.
- L270_275: the need to give a more detailed explanation of the method is obvious, since references to works [34,40,41] are not enough.
- L306: a short description of the Stabia thermal waters (temperature, discharge volume, chemical composition of water and gas) should be given. Link to publication 42 is not enough for the readers of the article.
- The material on hydrogeological conditions is insufficiently systematized. A description of the hydrogeological conditions is provided in the excerpts in Sections 2.3 and 4.5. A more complete and detailed description of the hydrogeological conditions of the area should have been provided.
The manuscript should be finalized for publication in the journal.
The article is undoubtedly interesting and relevant; contains new hydrogeochemical data for a complexly constructed aquifer in the urbanized territory of the Como region. Using traditional hydrogeochemical methods and the method of principal component analysis, a significant difference in the chemical composition of groundwater of the shallow and deep aquifers is shown. A conceptual model of the chemical composition of groundwater in the region is considered, the main role in which is played by the difference in the recharge areas of the shallow and deep aquifers. The authors also argue that the hydrogeochemical data used can serve as markers to characterize main factors affecting natural water quality, as well as a tool for recharge area reconstruction.
At the same time, the negligence in the design of the article materials gives rise to doubt in the constructions of the authors. The authors are invited to perform a more in-depth processing of hydrogeochemical materials and the text of the article. I hope that the comments below will help this.
- Firstly, the issue of anthropogenic pollution of groundwater remained outside the attention of the authors. The role of the anthropogenic load of the surface and ground waters under consideration has not been assessed. This is especially true of the shallow aquifer, for which there is an increased electrical conductivity and temperature (Fig. 3 a, b).
Without considering anthropogenic factor, the authors' constructions regarding the exclusive role of natural factors in explaining the peculiarities of the chemical composition of the shallow and deep aquifers are not convincing.
There is no data on contaminating components, except for NO3. This is a significant disadvantage of the work.
- Fig. 1 and Fig. 2a) should be edited to clearly indicate the spatial position and the location of territories in Fig. 1a, 1b and 2a as well as Fig. 7. As presented, it is not clear where the coring wells are in relation to the water sampling locations? Where the section A - A' (Fig. 7) is located in relation to Fig.1b?
All geographical objects mentioned in the text (rivers, streams, valleys, etc) should be shown in these figures.
- L257_L283 (3.3.1. Principal component analysis (PCA)): A more detailed explanation of the need to use principal component analysis in this study is needed. Links to publications 33,35,36 are not enough, explanations are needed for the readers of the article.
- L270_275: the need to give a more detailed explanation of the method is obvious, since references to works [34,40,41] are not enough.
- L306: a short description of the Stabia thermal waters (temperature, discharge volume, chemical composition of water and gas) should be given. Link to publication 42 is not enough for the readers of the article.
- The material on hydrogeological conditions is insufficiently systematized. A description of the hydrogeological conditions is provided in the excerpts in Sections 2.3 and 4.5. A more complete and detailed description of the hydrogeological conditions of the area should have been provided.
The manuscript should be finalized for publication in the journal.
The article is undoubtedly interesting and relevant; contains new hydrogeochemical data for a complexly constructed aquifer in the urbanized territory of the Como region. Using traditional hydrogeochemical methods and the method of principal component analysis, a significant difference in the chemical composition of groundwater of the shallow and deep aquifers is shown. A conceptual model of the chemical composition of groundwater in the region is considered, the main role in which is played by the difference in the recharge areas of the shallow and deep aquifers. The authors also argue that the hydrogeochemical data used can serve as markers to characterize main factors affecting natural water quality, as well as a tool for recharge area reconstruction.
At the same time, the negligence in the design of the article materials gives rise to doubt in the constructions of the authors. The authors are invited to perform a more in-depth processing of hydrogeochemical materials and the text of the article. I hope that the comments below will help this.
- Firstly, the issue of anthropogenic pollution of groundwater remained outside the attention of the authors. The role of the anthropogenic load of the surface and ground waters under consideration has not been assessed. This is especially true of the shallow aquifer, for which there is an increased electrical conductivity and temperature (Fig. 3 a, b).
Without considering anthropogenic factor, the authors' constructions regarding the exclusive role of natural factors in explaining the peculiarities of the chemical composition of the shallow and deep aquifers are not convincing.
There is no data on contaminating components, except for NO3. This is a significant disadvantage of the work.
- Fig. 1 and Fig. 2a) should be edited to clearly indicate the spatial position and the location of territories in Fig. 1a, 1b and 2a as well as Fig. 7. As presented, it is not clear where the coring wells are in relation to the water sampling locations? Where the section A - A' (Fig. 7) is located in relation to Fig.1b?
All geographical objects mentioned in the text (rivers, streams, valleys, etc) should be shown in these figures.
- L257_L283 (3.3.1. Principal component analysis (PCA)): A more detailed explanation of the need to use principal component analysis in this study is needed. Links to publications 33,35,36 are not enough, explanations are needed for the readers of the article.
- L270_275: the need to give a more detailed explanation of the method is obvious, since references to works [34,40,41] are not enough.
- L306: a short description of the Stabia thermal waters (temperature, discharge volume, chemical composition of water and gas) should be given. Link to publication 42 is not enough for the readers of the article.
- The material on hydrogeological conditions is insufficiently systematized. A description of the hydrogeological conditions is provided in the excerpts in Sections 2.3 and 4.5. A more complete and detailed description of the hydrogeological conditions of the area should have been provided.
The manuscript should be finalized for publication in the journal.
The article is undoubtedly interesting and relevant; contains new hydrogeochemical data for a complexly constructed aquifer in the urbanized territory of the Como region. Using traditional hydrogeochemical methods and the method of principal component analysis, a significant difference in the chemical composition of groundwater of the shallow and deep aquifers is shown. A conceptual model of the chemical composition of groundwater in the region is considered, the main role in which is played by the difference in the recharge areas of the shallow and deep aquifers. The authors also argue that the hydrogeochemical data used can serve as markers to characterize main factors affecting natural water quality, as well as a tool for recharge area reconstruction.
At the same time, the negligence in the design of the article materials gives rise to doubt in the constructions of the authors. The authors are invited to perform a more in-depth processing of hydrogeochemical materials and the text of the article. I hope that the comments below will help this.
- Firstly, the issue of anthropogenic pollution of groundwater remained outside the attention of the authors. The role of the anthropogenic load of the surface and ground waters under consideration has not been assessed. This is especially true of the shallow aquifer, for which there is an increased electrical conductivity and temperature (Fig. 3 a, b).
Without considering anthropogenic factor, the authors' constructions regarding the exclusive role of natural factors in explaining the peculiarities of the chemical composition of the shallow and deep aquifers are not convincing.
There is no data on contaminating components, except for NO3. This is a significant disadvantage of the work.
- Fig. 1 and Fig. 2a) should be edited to clearly indicate the spatial position and the location of territories in Fig. 1a, 1b and 2a as well as Fig. 7. As presented, it is not clear where the coring wells are in relation to the water sampling locations? Where the section A - A' (Fig. 7) is located in relation to Fig.1b?
All geographical objects mentioned in the text (rivers, streams, valleys, etc) should be shown in these figures.
- L257_L283 (3.3.1. Principal component analysis (PCA)): A more detailed explanation of the need to use principal component analysis in this study is needed. Links to publications 33,35,36 are not enough, explanations are needed for the readers of the article.
- L270_275: the need to give a more detailed explanation of the method is obvious, since references to works [34,40,41] are not enough.
- L306: a short description of the Stabia thermal waters (temperature, discharge volume, chemical composition of water and gas) should be given. Link to publication 42 is not enough for the readers of the article.
- The material on hydrogeological conditions is insufficiently systematized. A description of the hydrogeological conditions is provided in the excerpts in Sections 2.3 and 4.5. A more complete and detailed description of the hydrogeological conditions of the area should have been provided.
The manuscript should be finalized for publication in the journal.
The article is undoubtedly interesting and relevant; contains new hydrogeochemical data for a complexly constructed aquifer in the urbanized territory of the Como region. Using traditional hydrogeochemical methods and the method of principal component analysis, a significant difference in the chemical composition of groundwater of the shallow and deep aquifers is shown. A conceptual model of the chemical composition of groundwater in the region is considered, the main role in which is played by the difference in the recharge areas of the shallow and deep aquifers. The authors also argue that the hydrogeochemical data used can serve as markers to characterize main factors affecting natural water quality, as well as a tool for recharge area reconstruction.
At the same time, the negligence in the design of the article materials gives rise to doubt in the constructions of the authors. The authors are invited to perform a more in-depth processing of hydrogeochemical materials and the text of the article. I hope that the comments below will help this.
- Firstly, the issue of anthropogenic pollution of groundwater remained outside the attention of the authors. The role of the anthropogenic load of the surface and ground waters under consideration has not been assessed. This is especially true of the shallow aquifer, for which there is an increased electrical conductivity and temperature (Fig. 3 a, b).
Without considering anthropogenic factor, the authors' constructions regarding the exclusive role of natural factors in explaining the peculiarities of the chemical composition of the shallow and deep aquifers are not convincing.
There is no data on contaminating components, except for NO3. This is a significant disadvantage of the work.
- Fig. 1 and Fig. 2a) should be edited to clearly indicate the spatial position and the location of territories in Fig. 1a, 1b and 2a as well as Fig. 7. As presented, it is not clear where the coring wells are in relation to the water sampling locations? Where the section A - A' (Fig. 7) is located in relation to Fig.1b?
All geographical objects mentioned in the text (rivers, streams, valleys, etc) should be shown in these figures.
- L257_L283 (3.3.1. Principal component analysis (PCA)): A more detailed explanation of the need to use principal component analysis in this study is needed. Links to publications 33,35,36 are not enough, explanations are needed for the readers of the article.
- L270_275: the need to give a more detailed explanation of the method is obvious, since references to works [34,40,41] are not enough.
- L306: a short description of the Stabia thermal waters (temperature, discharge volume, chemical composition of water and gas) should be given. Link to publication 42 is not enough for the readers of the article.
- The material on hydrogeological conditions is insufficiently systematized. A description of the hydrogeological conditions is provided in the excerpts in Sections 2.3 and 4.5. A more complete and detailed description of the hydrogeological conditions of the area should have been provided.
The manuscript should be finalized for publication in the journal.
The article is undoubtedly interesting and relevant; contains new hydrogeochemical data for a complexly constructed aquifer in the urbanized territory of the Como region. Using traditional hydrogeochemical methods and the method of principal component analysis, a significant difference in the chemical composition of groundwater of the shallow and deep aquifers is shown. A conceptual model of the chemical composition of groundwater in the region is considered, the main role in which is played by the difference in the recharge areas of the shallow and deep aquifers. The authors also argue that the hydrogeochemical data used can serve as markers to characterize main factors affecting natural water quality, as well as a tool for recharge area reconstruction.
At the same time, the negligence in the design of the article materials gives rise to doubt in the constructions of the authors. The authors are invited to perform a more in-depth processing of hydrogeochemical materials and the text of the article. I hope that the comments below will help this.
- Firstly, the issue of anthropogenic pollution of groundwater remained outside the attention of the authors. The role of the anthropogenic load of the surface and ground waters under consideration has not been assessed. This is especially true of the shallow aquifer, for which there is an increased electrical conductivity and temperature (Fig. 3 a, b).
Without considering anthropogenic factor, the authors' constructions regarding the exclusive role of natural factors in explaining the peculiarities of the chemical composition of the shallow and deep aquifers are not convincing.
There is no data on contaminating components, except for NO3. This is a significant disadvantage of the work.
- Fig. 1 and Fig. 2a) should be edited to clearly indicate the spatial position and the location of territories in Fig. 1a, 1b and 2a as well as Fig. 7. As presented, it is not clear where the coring wells are in relation to the water sampling locations? Where the section A - A' (Fig. 7) is located in relation to Fig.1b?
All geographical objects mentioned in the text (rivers, streams, valleys, etc) should be shown in these figures.
- L257_L283 (3.3.1. Principal component analysis (PCA)): A more detailed explanation of the need to use principal component analysis in this study is needed. Links to publications 33,35,36 are not enough, explanations are needed for the readers of the article.
- L270_275: the need to give a more detailed explanation of the method is obvious, since references to works [34,40,41] are not enough.
- L306: a short description of the Stabia thermal waters (temperature, discharge volume, chemical composition of water and gas) should be given. Link to publication 42 is not enough for the readers of the article.
- The material on hydrogeological conditions is insufficiently systematized. A description of the hydrogeological conditions is provided in the excerpts in Sections 2.3 and 4.5. A more complete and detailed description of the hydrogeological conditions of the area should have been provided.
The manuscript should be finalized for publication in the journal.
The article is undoubtedly interesting and relevant; contains new hydrogeochemical data for a complexly constructed aquifer in the urbanized territory of the Como region. Using traditional hydrogeochemical methods and the method of principal component analysis, a significant difference in the chemical composition of groundwater of the shallow and deep aquifers is shown. A conceptual model of the chemical composition of groundwater in the region is considered, the main role in which is played by the difference in the recharge areas of the shallow and deep aquifers. The authors also argue that the hydrogeochemical data used can serve as markers to characterize main factors affecting natural water quality, as well as a tool for recharge area reconstruction.
At the same time, the negligence in the design of the article materials gives rise to doubt in the constructions of the authors. The authors are invited to perform a more in-depth processing of hydrogeochemical materials and the text of the article. I hope that the comments below will help this.
- Firstly, the issue of anthropogenic pollution of groundwater remained outside the attention of the authors. The role of the anthropogenic load of the surface and ground waters under consideration has not been assessed. This is especially true of the shallow aquifer, for which there is an increased electrical conductivity and temperature (Fig. 3 a, b).
Without considering anthropogenic factor, the authors' constructions regarding the exclusive role of natural factors in explaining the peculiarities of the chemical composition of the shallow and deep aquifers are not convincing.
There is no data on contaminating components, except for NO3. This is a significant disadvantage of the work.
- Fig. 1 and Fig. 2a) should be edited to clearly indicate the spatial position and the location of territories in Fig. 1a, 1b and 2a as well as Fig. 7. As presented, it is not clear where the coring wells are in relation to the water sampling locations? Where the section A - A' (Fig. 7) is located in relation to Fig.1b?
All geographical objects mentioned in the text (rivers, streams, valleys, etc) should be shown in these figures.
- L257_L283 (3.3.1. Principal component analysis (PCA)): A more detailed explanation of the need to use principal component analysis in this study is needed. Links to publications 33,35,36 are not enough, explanations are needed for the readers of the article.
- L270_275: the need to give a more detailed explanation of the method is obvious, since references to works [34,40,41] are not enough.
- L306: a short description of the Stabia thermal waters (temperature, discharge volume, chemical composition of water and gas) should be given. Link to publication 42 is not enough for the readers of the article.
- The material on hydrogeological conditions is insufficiently systematized. A description of the hydrogeological conditions is provided in the excerpts in Sections 2.3 and 4.5. A more complete and detailed description of the hydrogeological conditions of the area should have been provided.
The manuscript should be finalized for publication in the journal.
The article is undoubtedly interesting and relevant; contains new hydrogeochemical data for a complexly constructed aquifer in the urbanized territory of the Como region. Using traditional hydrogeochemical methods and the method of principal component analysis, a significant difference in the chemical composition of groundwater of the shallow and deep aquifers is shown. A conceptual model of the chemical composition of groundwater in the region is considered, the main role in which is played by the difference in the recharge areas of the shallow and deep aquifers. The authors also argue that the hydrogeochemical data used can serve as markers to characterize main factors affecting natural water quality, as well as a tool for recharge area reconstruction.
At the same time, the negligence in the design of the article materials gives rise to doubt in the constructions of the authors. The authors are invited to perform a more in-depth processing of hydrogeochemical materials and the text of the article. I hope that the comments below will help this.
- Firstly, the issue of anthropogenic pollution of groundwater remained outside the attention of the authors. The role of the anthropogenic load of the surface and ground waters under consideration has not been assessed. This is especially true of the shallow aquifer, for which there is an increased electrical conductivity and temperature (Fig. 3 a, b).
Without considering anthropogenic factor, the authors' constructions regarding the exclusive role of natural factors in explaining the peculiarities of the chemical composition of the shallow and deep aquifers are not convincing.
There is no data on contaminating components, except for NO3. This is a significant disadvantage of the work.
- Fig. 1 and Fig. 2a) should be edited to clearly indicate the spatial position and the location of territories in Fig. 1a, 1b and 2a as well as Fig. 7. As presented, it is not clear where the coring wells are in relation to the water sampling locations? Where the section A - A' (Fig. 7) is located in relation to Fig.1b?
All geographical objects mentioned in the text (rivers, streams, valleys, etc) should be shown in these figures.
- L257_L283 (3.3.1. Principal component analysis (PCA)): A more detailed explanation of the need to use principal component analysis in this study is needed. Links to publications 33,35,36 are not enough, explanations are needed for the readers of the article.
- L270_275: the need to give a more detailed explanation of the method is obvious, since references to works [34,40,41] are not enough.
- L306: a short description of the Stabia thermal waters (temperature, discharge volume, chemical composition of water and gas) should be given. Link to publication 42 is not enough for the readers of the article.
- The material on hydrogeological conditions is insufficiently systematized. A description of the hydrogeological conditions is provided in the excerpts in Sections 2.3 and 4.5. A more complete and detailed description of the hydrogeological conditions of the area should have been provided.
The manuscript should be finalized for publication in the journal.
Author Response
The article is undoubtedly interesting and relevant; contains new hydrogeochemical data for a complexly constructed aquifer in the urbanized territory of the Como region. Using traditional hydrogeochemical methods and the method of principal component analysis, a significant difference in the chemical composition of groundwater of the shallow and deep aquifers is shown. A conceptual model of the chemical composition of groundwater in the region is considered, the main role in which is played by the difference in the recharge areas of the shallow and deep aquifers. The authors also argue that the hydrogeochemical data used can serve as markers to characterize main factors affecting natural water quality, as well as a tool for recharge area reconstruction.
At the same time, the negligence in the design of the article materials gives rise to doubt in the constructions of the authors. The authors are invited to perform a more in-depth processing of hydrogeochemical materials and the text of the article. I hope that the comments below will help this.
We thank the reviewer for the helpful suggestions and the exertion in reviewing our manuscript. We answered point by point to the comments in the following paragraphs and we reviewed the text of the manuscript according to the reviewer’s comments. We also kindly note that all the text lines indicated in the responses refer to the “marked change” version of the revised manuscript.
- Firstly, the issue of anthropogenic pollution of groundwater remained outside the attention of the authors. The role of the anthropogenic load of the surface and ground waters under consideration has not been assessed. This is especially true of the shallow aquifer, for which there is an increased electrical conductivity and temperature (Fig. 3 a, b).
Without considering anthropogenic factor, the authors' constructions regarding the exclusive role of natural factors in explaining the peculiarities of the chemical composition of the shallow and deep aquifers are not convincing.
There is no data on contaminating components, except for NO3. This is a significant disadvantage of the work.
In this paper, elements possibly sourcing from natural phenomena (i.e., rock-water and sediment-water interaction) in analyzed waters were mainly discussed in order to better clarify the natural tracers for the reconstruction of recharge areas and of the hydrogeological conceptual model. NO3 was in fact added to the chemical variables considering its natural input from nutrients in surface waters. However, also some potentially anthropogenic trace elements (i.e., Cd, Pb and Ag) were analyzed in water samples by ICP-MS. These elements concentration is below the LOD in most of the samples (more than 50%) and all the samples generally have a very low concentration. Considering the reviewer’s comment, we decided to add these elements values in Table S3 anyway, as well as a short description in text (in lines 375-378 and 568) confirming that the main processes driving the chemistry of water in these aquifers are natural. We believe that these evidences better define the geochemical anomaly of As too, and we thank the reviewer for raising this issue.
- Fig. 1 and Fig. 2a) should be edited to clearly indicate the spatial position and the location of territories in Fig. 1a, 1b and 2a as well as Fig. 7. As presented, it is not clear where the coring wells are in relation to the water sampling locations? Where the section A - A' (Fig. 7) is located in relation to Fig.1b?
All geographical objects mentioned in the text (rivers, streams, valleys, etc) should be shown in these figures.
Following reviewer's comment, we edited the Figure 1 adding a frame of Figure 2a setting and the trace of Figure7 section. Moreover, we indicated in the caption the localities of the two cores (which are coincident with two water sampling points). We also added more labels to indicate all the geographical features indicated in text.
- L257_L283 (3.3.1. Principal component analysis (PCA)): A more detailed explanation of the need to use principal component analysis in this study is needed. Links to publications 33,35,36 are not enough, explanations are needed for the readers of the article.
We thank the reviewer for this observation. We added more detail regarding the usefulness of Principal component analysis for environmental chemical data interpretation and more references in lines 279-284.
- L270_275: the need to give a more detailed explanation of the method is obvious, since references to works [34,40,41] are not enough.
We added a sentence in lines 290-292 and more references to justify and explain this approach according to reviewer’s suggestion.
- L306: a short description of the Stabia thermal waters (temperature, discharge volume, chemical composition of water and gas) should be given. Link to publication 42 is not enough for the readers of the article.
According to reviewer’s comment, we added a small description of Stabio waters where first mentioned, in the section 2.3, lines 110-113.
- The material on hydrogeological conditions is insufficiently systematized. A description of the hydrogeological conditions is provided in the excerpts in Sections 2.3 and 4.5. A more complete and detailed description of the hydrogeological conditions of the area should have been provided.
The manuscript should be finalized for publication in the journal.
We appreciate the comment of the reviewer and we tried to define at best the hydrogeological setting of the study area. The local hydrogeological setting has been the focus of scientific efforts since the mid-1970s; most of the information derives from grey literature (municipal reports, private well logs) and is summarized in a few publications (references 14, 15, 17, 26, 32, 33 in the paper) and unpublished thesis. Nevertheless, large unknowns still persist in our knowledge of the local hydrogeology, due to a number of reasons:
- Most of the studies focused on specific, applied goals: for instance, in 1976 a municipal commission has been established to study the subsidence phenomena affecting the urban area; the commission results were published in 1980 and identified groundwater extraction as the main cause of subsidence. Other studies were conducted since 1998 in relation to an engineering project of moveable bulkheads, but focused on the lakeshore area only.
- Regarding the Deep aquifer, in the late 1970s studies included monitoring of the water level in this aquifer. at that time, active pumping was ongoing and the aquifer was thus artificially depressed, while nowadays pumping is forbidden and the deep aquifer recovered its head. Measurements collected by the University of Insubria team in 2014 (unpublished data, which we mention in the paper at lines 512-514) have shown an overpressure hydraulic head, but the number of measurement points and the frequency of measures does not allow to properly reconstruct the deep aquifer circulation.
- To our knowledge, the geochemical monitoring realized in this study is the first of such a kind on a basin scale and provide original data which implement the available knowledge.
As suggested by the reviewer, we tried to emphasize the knowledge gaps in the revised version of the text, in lines 182-183 and line 189.